# Effects of Rossby Waves Breaking and Atmospheric Blocking Formation on the Extreme Forest Fire and Floods in Eastern Siberia 2019

Olga Yu. Antokhina [1], Pavel N. Antokhin [1], Boris D. Belan [1], Alexander V. Gochakov [1,2],
Yuliya V. Martynova [3], Konstantin N. Pustovalov [3,4], Lena D. Tarabukina [5] and Elena V. Devyatova [6,*]

1 V.E. Zuev Institute of Atmospheric Optics of SB RAS, Tomsk 663055, Russia; antokhina@iao.ru (O.Y.A.)
2 Siberian Regional Hydrometeorological Research Institute, Novosibirsk 630099, Russia
3 Institute of Monitoring of Climatic and Ecological Systems of SB RAS, Tomsk 663055, Russia
4 Department of Meteorology and Climatology, National Research Tomsk State University, Tomsk 634028, Russia
5 Yu.G. Shafer Institute of Cosmophysical Research and Aeronomy, SB RAS, Yakutsk 677980, Russia
6 Institute of Solar-Terrestrial Physics of SB RAS, Irkutsk 664033, Russia
* Correspondence: devyatova@iszf.irk.ru; Tel.: +7-914-876-2328

**Abstract:** In 2019, the southern region of Eastern Siberia (located between 45° N and 60° N) experienced heavy floods, while the northern region (between 60° N and 75° N) saw intense forest fires that lasted for almost the entire summer, from 25 June to 12 August. To investigate the causes of these natural disasters, we analyzed the large-scale features of atmospheric circulation, specifically the Rossby wave breaking and atmospheric blocking events. In the summer of 2019, two types of Rossby wave breaking were observed: a cyclonic type, with a wave breaking over Siberia from the east (110° E–115° E), and an anticyclonic type, with a wave breaking over Siberia from the west (75° E–90° E). The sequence of the Rossby wave breaking and extreme weather events in summer, 2019 are as follows: 24–26 June (cyclonic type, extreme precipitation, flood), 28–29 June and 1–2 July (anticyclonic type, forest fires), 14–17 July (both types of breaking, forest fires), 25–28 July (cyclonic type, extreme precipitation, flood), 2 and 7 August (anticyclonic type, forest fires). Rossby wave breaking occurred three times, resulting in the formation and maintenance of atmospheric blocking over Eastern Siberia: 26 June–3 July, 12–21 July and 4–10 August. In general, the scenario of the summer events was as follows: cyclonic Rossby wave breaking over the southern part of Eastern Siberia (45° N–60° N) caused extreme precipitation (floods) and led to low gradients of potential vorticity and potential temperature in the west and east of Lake Baikal. The increased wave activity flux from the Europe–North Atlantic sector caused the anticyclonic-type Rossby wave breaking to occur west of the area of a low potential vorticity gradient and north of 60° N. This, in turn, contributed to the maintenance of blocking anticyclones in the north of Eastern Siberia, which led to the intensification and expansion of the area of forest fires. These events were preceded by an increase in the amplitude of the quasi-stationary wave structure over the North Atlantic and Europe during the first half of June.

**Keywords:** forest fires; precipitation; Siberia; Rossby wave breaking; atmospheric blocking; wave activity flux; temperature





## 1. Introduction

The increase in weather extremes remains a crucial question linked to global climate change [1,2]. The air temperature and precipitation are among the most critical climate indicators, and they are closely linked to extreme weather events such as droughts, wildfires, and floods during boreal summer. According to research conducted by Groisman et al. in 2017 [2], numerous studies have indicated a rise in precipitation intensity in Northern

Eurasia. Additionally, extended periods of no rainfall have been accompanied by summer droughts and an increase in unusual temperature patterns. These changes have led to an increase in the occurrence of forest fires and floods [3,4]. Both fires and floods are extremely dangerous for the Russian economy and human health.

Furthermore, forest fires may affect regional air quality and human health and feedback processes between the climate and the biosphere due to the emission of atmospheric carbon dioxide and aerosols [5–7]. In the past, the extent of Siberia's boreal fires was underestimated in terms of their contribution to global fire emissions. Currently, the Siberian forest fires have been the focus of many scientific papers, recognized as one of the most dramatic phenomena. Soja et al. 2004 [6] showed that boreal fire is significant to the global carbon budget. Siberia's boreal forests are estimated to contain two-thirds of the world's total boreal forests [5]. According to [3], more than 70% and up to 90% of the total area burned in Russia occurs in Siberia. The Eastern Siberian and Far Eastern regions are particularly prone to intense forest fires [8], which can be hazardous due to their contribution to Arctic ice melting through the settling of black carbon [9].

In the summer of 2019, Eastern Siberia (ES) experienced record-breaking floods with peaks at the end of June and the end of July, caused by extreme rainfall [10,11], https://tass.com/floods-in-irkutsk-region, accessed on 11 March 2023, and extreme, long-duration forest fires from the end of June to mid-August caused by dry thunderstorms and high air temperature [https://tass.ru/proisshestviya/6703544, accessed on 11 March 2023, https://www.nasa.gov/image-feature/goddard/2019/huge-wildfires-in-russias-siberian-province-continue, accessed on 11 March 2023]. The search for the large- and synoptic-scale atmospheric conditions that contributed to the occurrence of both forest fires and floods in Siberia in 2019 is crucial in order to better understand and potentially predict such weather extremes. Many scientific studies have shown that extreme weather events are caused by upper tropospheric ridges and troughs associated with the propagation, stationarity, and breaking of Rossby waves [12–19]. According to the findings of Moore et al. 2019 [20] and Liu 2017 [21], Rossby wave breaking (RWB) plays a crucial role in both horizontal and vertical large-scale transport and mixing, as demonstrated in observations and idealized general circulation models. The most extraordinary example demonstrating how the same large-scale event (RWB) caused both wildfire and flood is the atmospheric pattern in July–August 2010, the Rossby wave breaking caused the Russian heatwave (fire) and Pakistan flood [22]. Of course, high summer rainfall and floods in continental extratropics are frequently related to regional convective storms [23]. However, regional storms are often also forced by synoptic- and large-scale atmospheric dynamics. Thus, in papers [24,25] the importance of cyclones and their associated frontal systems, for the occurrence of regional-scale precipitation extremes is quantified. The current state of knowledge regarding large-scale meteorological patterns associated with short-duration (less than 1 week) extreme precipitation events over North America is considered in [26]. Bosart and Moore, 2017 [27] highlight how the large- and synoptic-scale flow can evolve to facilitate multiple geographically separated but dynamically linked extreme weather events in North America in October 2007. The study presented in [28] analyzed precipitation events in the Selenga river basin and atmospheric blocking over Eurasia during July from 1979 to 2017. The results showed that when there was joint blocking over Europe and the Russian Far East (RFE), it led to aridity over the southern (Mongolian) part of the Selenga basin and increased precipitation over the northern (Russian) part of the basin.

Li and Ruan, 2018 [29] and Xu et al., 2019 [30] suggested two teleconnections over northern Eurasia between North Atlantic and the Eurasian continent in the summertime. The first pattern termed the Atlantic–Eurasian (AEA) teleconnection [29] has five action centers in the middle troposphere: subtropical North Atlantic Ocean, Eastern Europe, Mongolia–north China, the northeastern North Atlantic Ocean, and the Kara Sea–Northern Siberia. According to Li and Ruan (2018) [29], the AEA is a large-scale Rossby wave train that originates in the subtropical North Atlantic Ocean. The second pattern, the

British–Baikal Corridor (BBC), proposed in [30], consists of four geographically fixed centers located in the upper troposphere: over the west of the British Isles, the Baltic Sea, western Siberia, and Lake Baikal. The authors of [30] suggest that the BBC pattern is related to the summertime upper-tropospheric polar front jet. Here, we focus on these patterns as potential sources of wave energy that may contribute to the breaking of Rossby waves and the formation of blockings over Siberia. The variability of the Asian summer monsoon anticyclone (ASMA) has been a recent focus of research, as reported in [31]. The ASMA is located between the subtropical westerly jet to the north and easterly jets to the south. Therefore, it can be viewed as a potential source of excitation or modulation of Rossby waves, which could impact the atmospheric circulation over Siberia during the summer season.

The changes in the trend of RWB in the Northern hemisphere during the last four decades were revealed by Bowley et al. 2019 [32] and Jing and Banerjee, 2018 [33]. Jing and Banerjee, 2018 [33] discovered that the frequency of both types of breaking (AWB and CWB) has increased since 1981 above 320 K, and their mean latitude has shifted poleward. They also revealed that such changes in AWB frequency and latitudinal area are more significant in summer than in winter [33]. Bowley and his co-authors in their work [32] suggest that the increase in the frequency of AWB in summer is likely due to the Asian monsoon. The identified trends in RWB may indicate a more frequent occurrence of blockings in Siberia during the summer, and a poleward shift of these events.

The RWBs are typically studied using isentropic potential vorticity (PV) as a diagnostic tool [34–36]. The rapid and irreversible deformation of PV contours is observed during the amplification and breaking of Rossby waves [37]. The RWB is often identified by the reversal of the latitudinal gradient of PV on isentropic surfaces (or potential temperature on the dynamical tropopause—PV-Θ) [38,39]. According to the direction of PV/PV-Θ contour deformation, the types of RWB are divided into cyclonic and anticyclonic (CWB and AWB) [21,37,40,41]. Several studies have demonstrated that RWB plays a crucial role in the formation and persistence of blocking patterns, including in the Siberian region [15,38,42,43]. Additionally, Chyi et al. 2019 [43] reported that the frequency of blocking and RWB in Siberia is high during summer, particularly in late July. This phenomenon can be explained by the "Northward jump of the Asian jet stream" in summer [44].

It is known that midlatitude circulation is predominant for the boreal forest area. In turn, mid-latitude circulation is characterized by the strong dependence of surface temperature on circulation patterns (cyclones, anticyclones, blocking, high amplitude ridges, and troughs). In discussions related to heatwaves and long-lasting droughts in mid-latitude regions, particular attention is paid to the propagation of Rossby waves (RWP) and their breaking, as well as atmospheric blocking (AB) in the middle and upper troposphere [13,17,45–48]. The positive feedback between heat waves associated with RW/AB and soil moisture has also been discussed [13,49,50]. The effect of RW on forest fires in both the northern and southern hemispheres was discussed by Hayasaka et al. 2019 [51] and Reeder et al. 2015 [52]. Both papers have concluded that forest fire occurrences are associated with the presence of warm and dry air masses, which can be facilitated by the propagation of the Rossby waves [52] and the meandering of westerly flow [51].

Chyi et al. 2019 [43] showed that both AWB and CWB types are associated with precipitation in Central Siberia. They demonstrated the dynamic processes for AWB and CWB events and how they lead to different precipitation patterns in the region. The deepening of the trough from the sub-Arctic region was found to be associated with both types of breaking by Chyi et al. 2019 [43]. Antokhina et al. 2019 [53] found a statistically significant relationship between blocking frequency and precipitation in Siberia during July. It has been shown that the north–south precipitation anomalies dipole is associated with atmospheric blocking. It was explained by the RWB direction and the blocking formation. During RWB, cold air masses (high PV) are advected to the south, while warm (low PV) air masses are advected to the north. Hence, the RWB and blocking formation can promote

the stable vertical anticyclone structure in north Siberia (forest fire) and the unstable cutoff cyclone structure in the south (flood). The process associated with extreme precipitation is also related to the mixing of air masses from the high-latitude regions of the stratosphere into the low-latitude regions of the troposphere and vice versa, due to RWB [34,54].

The aim of this work is to conduct a process-oriented analysis and evaluation of the Rossby wave propagation, their breaking, blocking formation, floods, and forest fires in Siberia during the summer of 2019. First, we present the chronology for all processes in June–July–August 2019. Second, we estimate air masses' properties (horizontal exchange) and dynamical tropopause properties (instability, vertical mixing) for RWB events.

## 2. Data and Methods

### 2.1. Data

The scale and intensity of forest fires, as well as the background air quality, were estimated using carbon monoxide emissions from biomass burning, wildfire hotspot data, aerosol concentrations recorded from 1 June to 31 August 2019.

Carbon monoxide emission (CObb) from biomass burning data was obtained from the global fire assimilation system (GFAS) [55] (CAMS global fire assimilation system: http://apps.ecmwf.int/datasets/data/cams-gfas/, accessed on 12 December 2021). GFAS is based on the fire radiative power (FRP) from the MODIS instrument onboard the National Aeronautics and Space Administration (NASA) Terra and Aqua satellites (daily averaged FRP with 0.1° resolution) [55]. The clustering procedure was applied to gridded GFAS data. The DBSCAN (density-based spatial clustering of applications with noise) clustering algorithm was used [56]. We use the following parameters: Eps = 0.6 grad. ("the maximum distance between two samples for one to be considered as in the neighborhood of the other", https://scikit-learn.org, accessed on 12 December 2021) and MinPts = 7 ("the number of samples (or total weight) in a neighborhood for a point to be considered as a core point. This includes the point itself", https://scikit-learn.org, accessed on 12 December 2021).

For analyzing wildfire hotspots, the fire information for resource management system (FIRMS) (https://modaps.modaps.eosdis.nasa.gov/services/about/products/c6-nrt/MOD14.html, accessed on 12 December 2021) and the hotspots visualizer Worldview based on satellite images (http://worldview.earthdata.nasa.gov, accessed on 12 December 2021) were used. The hotspots data in FIRMS [57,58] are based on "Fires and Thermal Anomalies" (MOD14/MYD14) obtained by the spectroradiometer MODIS on Channels 4 and 11 μm from the Terra and Aqua satellites. The algorithm for hotspot detection is based on recognizing thermal anomalies on the surface.

To demonstrate the effect of forest fires on the background air quality, we utilized aerosol data with a diameter of 0.25 μm, which was derived from the Fonovaya station [59], http://lop.iao.ru/EN/fon/diffbat/, accessed on 12 December 2021, https://peexhq.home.blog/2019/09/13/siberian-aerosol-measurements-at-fonovaya-station/, accessed on 12 December 2021 located in the border between Western and Eastern Siberia.

For the analysis of the daily number of lightning strokes, we used data obtained from the Worldwide Lightning Location Network [60], WWLLN, http://wwlln.net/, accessed on 12 December 2021 with horizontal resolution 0.25° × 0.25°. The WWLLN records approximately 10–20% of lightning strokes that have the highest charge, which are often associated with positive "cloud-ground" lightning, including so-called "dry lightning". Therefore, for the study of forest fires, the WWLLN data is appropriate for lightning analysis. For precipitation analysis, we used the first guess daily product with a spatial resolution of 1° from the global precipitation climatology centre (Deutscher Wetterdienst) (GPCC) [61], https://opendata.dwd.de/climate_environment/GPCC/html/gpcc_firstguess_daily_doi_download.html, accessed on 12 December 2021 from 1 June to 31 August. The GPCC first guess is particularly useful for monitoring extreme weather events [61,62]. This set represents ground observations of daily precipitation derived from the quality-controlled stations. This was important for the study because the paper focuses

on extreme precipitation. The GPCC dataset is in good agreement with the weather station data for the Siberia area [63].

Four synoptic hour (0, 6, 12, 18 UTC) data (potential temperature at the 2 potential vorticity units (PVU) level (dynamic tropopause), potential vorticity at 350 K level, geopotential height, temperature, u,v wind components at 500 hPa, 10 m temperature and u,v wind components, convective available potential energy and total column water) from the ECMWF ERA-Interim reanalysis datasets [64], https://apps.ecmwf.int/datasets/data/interim-full-daily/levtype=sfc/, accessed on 12 December 2021 were used in this study for 1 June–31 August 2019. The horizontal resolution is $1.5° \times 1.5°$ and $2.5° \times 2.5°$.

### 2.2. Atmospheric Blocking (AB, Blocks)

There are several ways to detect blocking. We use the blocking index as the Barriopedro et al. 2006 [65]. This index is based on the *GHGS* criterion proposed by Lejenäs and Økland 1980 [66] and the *GHGN* criterion suggested by Tibaldi and Molteni 1991 [67]. The 500 hPa geopotential height gradients (*GHG*) north and south (*GHGN* and *GHGS*, respectively) are calculated. Barriopedro et al. 2006 [65] used the five values of Δ.

$$GHGS = \frac{Z(\varphi_0) - Z(\varphi_s)}{\varphi_0 - \varphi_s} \tag{1}$$

$$GHGN = \frac{Z(\varphi_n) - Z(\varphi_0)}{\varphi_n - \varphi_0} \tag{2}$$

$Z$ − 500 hPa geopotential height, $\varphi_n = 77.5° \, N + \Delta$, $\varphi_0 = 60° \, N + \Delta$, $\varphi_s = 40° \, N \pm \Delta$, $\Delta = -5.0°, -2.5°, 0°, 2.5°$ or $5.0°$.

A longitude is considered blocked when $GHGS > 0$, $GHGN < 10$ m/deg for at least one of the five Δ values.

### 2.3. Rossby Wave Breaking (RWB) and Wave Activity Flux (WAF)

The detection of RWB was performed by using isentropic potential vorticity (PV) [37]. RWB are characterized by a poleward intrusion of low potential vorticity (or high potential temperature) air and an equatorward intrusion of high potential vorticity (or low potential temperature) [68]. The detection of RWB events in this paper is based on the overturning contour identification technique developed by Strong and Magnusdottir, 2008 [40]. The overturning technique can be applied to contour on the dynamical tropopause (DT), isentropic surface (Θ), or pressure surface. For DT and Θ, the techniques are dynamically consistent; low potential temperature streamers on the DT are generally equivalent to high potential vorticity streamers on isentropic surfaces. Jing and Banerjee, 2018 used the isentropic surfaces 320, 350, and 370 K. Bowley et al. 2019 [32] used DT (potential temperature on the DT–PV-Θ). It was discovered that the area with a high frequency of CWB is located eastward of Lake Baikal.

We applied the identification and analysis of RWB the following way:

1. For the isentropic surface at 350 K (which is used to reveal the exchange along the subtropical tropopause [37]), an automatic algorithm was used to search for the overturning contour from 1 to 9 PVU with an interval of 0.5 PVU. For the automatic detection of centers and squares of overturning areas, we used the identification technique developed by Barnes and Hartmann, 2012 [39];
2. For each day between 1 June and August 31, we conducted a synoptic analysis of the potential temperature on the DT (PV-Θ) maps. We utilized PV-Θ to analyze air masses transformation and blocking formation in the mid-latitude area [42,43];
3. For a 3D visualization of RWB processes, we calculated the 3D surface of DT (2PVU) in three dimensions: longitude, latitude, and geopotential height.

The wave activity flux (WAF) that indicates the propagation of planetary waves can usually be used to localize regions of wave activity sources and sinks. Wave energy

propagation is described by the horizontal wave activity flux (WAF) at 250 hPa. The wave activity flux proposed by Plumb in 1985 is utilized in this study [69].

## 3. Results

### 3.1. The Scenario RWB, Blocks, Precipitation, and Forest Fires with a Synoptic View

In order to conduct a process-oriented analysis and evaluation of the Rossby wave propagation, their breaking, blocking formation, floods, and forest fires in Siberia during the summer of 2019 we started our study by tracing the chronology for all the characteristics we have that describe the event. The results are shown in Figures 1–3 and in Table S1 (in Supplementary Materials).

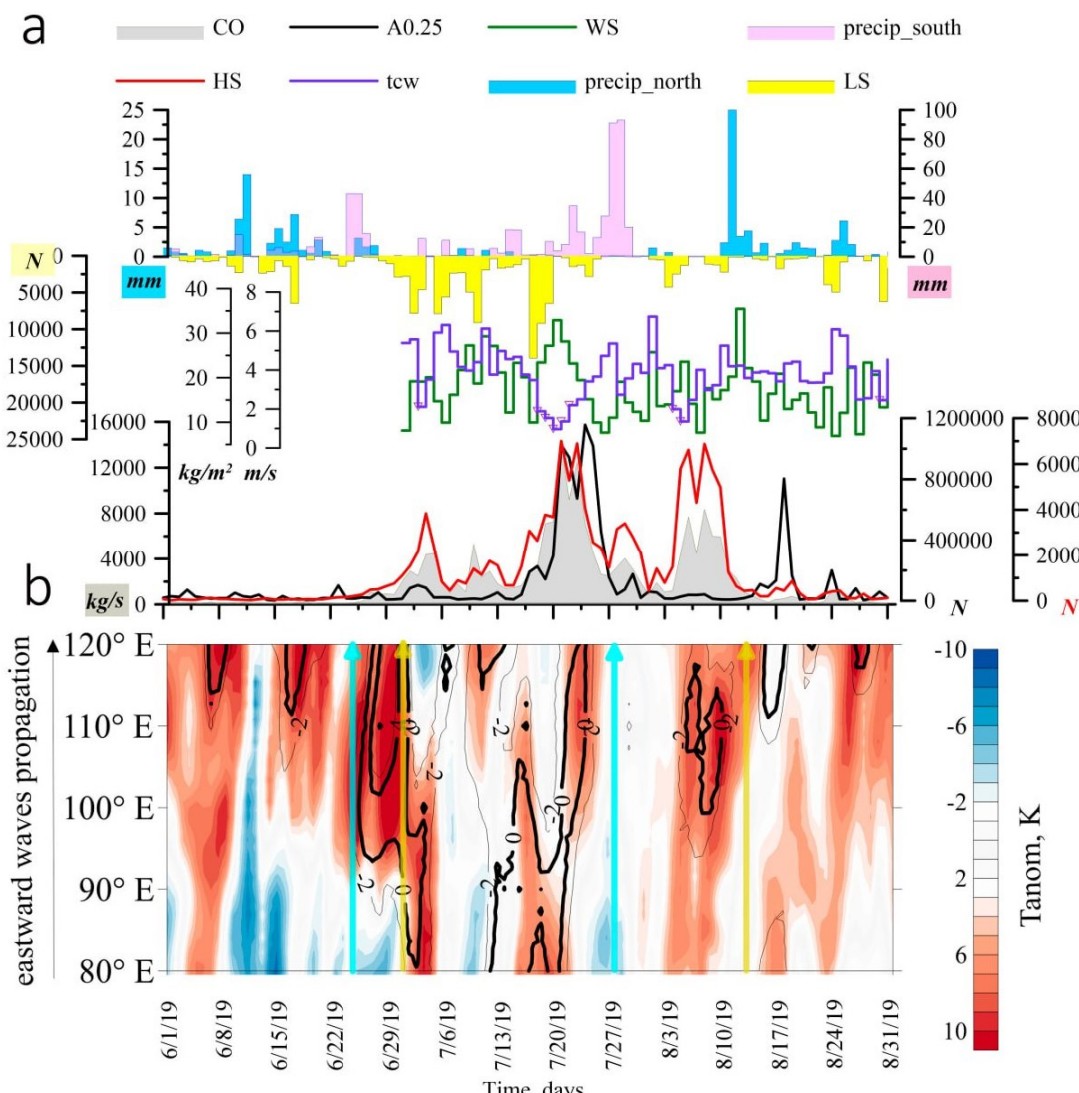

**Figure 1.** (**a**) Total CO emission (90° E–120° E, 55° N–65° N) (kg/s), A0.25 aerosol concertation in Western Siberia (Fonovaya) (N), HS—the total number of hotspots (90° E–120° E, 55° N–65° N); the total number of the lightning strikes (LS); WS—surface wind speed; tcw—total column water; precip—atmospheric precipitation. tcw, ws, precip_north for the center of forest fire area 105° E–60° N, precip south: for June 54.5° N–97.5° E, for July the average amount for two grid points: 51.5° N, 103.5° E and 104.5° E; (**b**) Time-longitude cross-sections of *GHGS* (m/°lat). *GHGS* > 0 corresponds blocking, *GHGS* > −2—near blocking, Era-Interim data, red and blue fill—surface temperatures anomalies. Light blue vertical line—high precipitation events; yellow vertical line—start and finish of forest fire period.

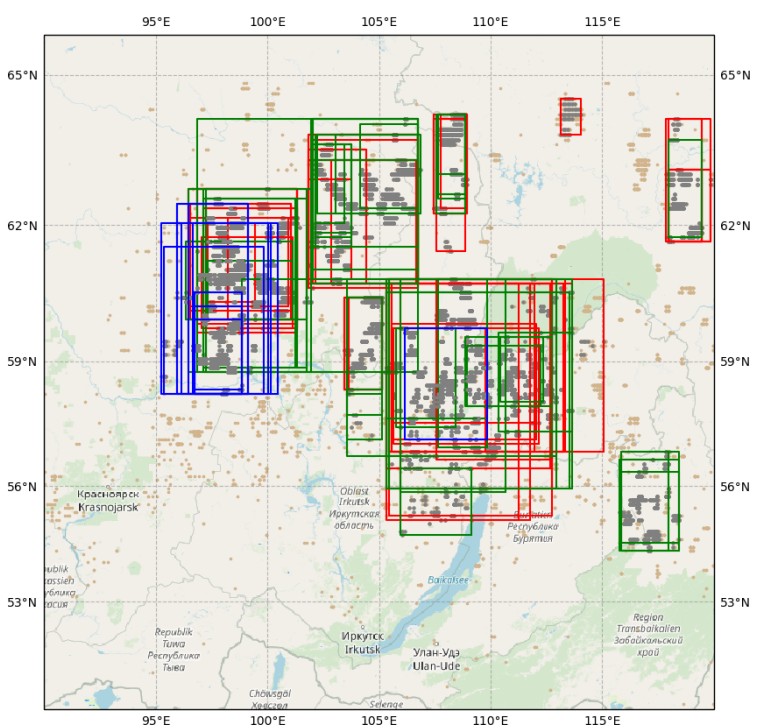

**Figure 2.** The forest fire clusters are obtained using the DBSCAN algorithm based on GFAS data from 1 June–31 August. Blue—groups existed from 1 to 15 July; red—16–31 July; green—1–12 August. Area 1st—58° N–63° N, 95° E–102° E, 2nd—56° N–61° N, 104° E–115° E, 3rd—61° N–64° N, 102° E–108° E.

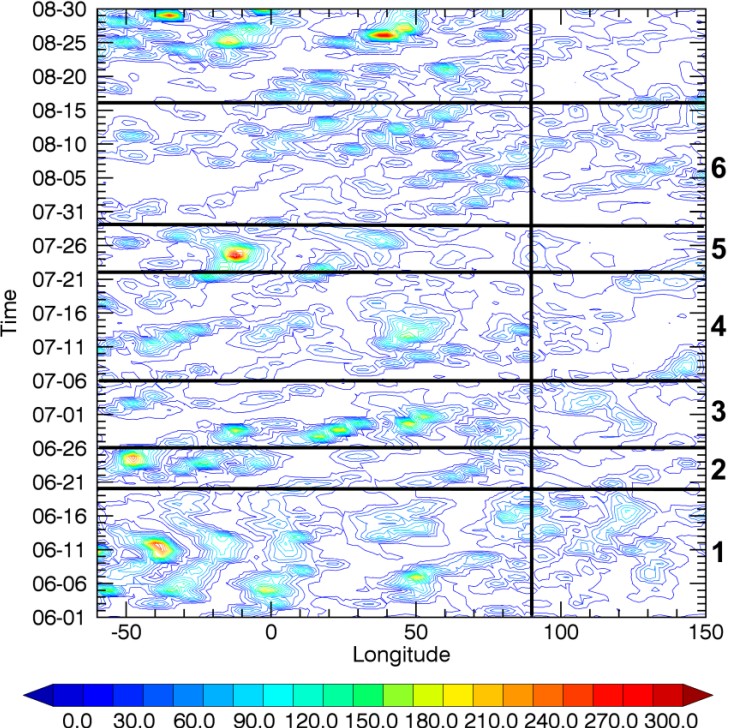

**Figure 3.** Horizontal (S) component of the WAF (m² s⁻²), according to Plumb equation (1985) for 50° N–70° N. The black vertical line is the western border of Eastern Siberia. Black horizontal lines are six-time interval boundaries: 1—preliminary period; 2—the first flood; 3—first forest fire large cluster formation; 4—second and third forest fire cluster formation and forest fire amplification; 5—the second flood; 6—forest fire period, amplification forest fires activity. Era-Interim data.

Figure 1 provides visual representations of the dynamics of forest fire intensity, precipitation, and atmospheric circulation characteristics during the summer of 2019. Specifically, Figure 1a includes the following graphs:

(1)    Bottom graphs: the day-to-day variation of total CO emission (90° E–120° E, 55° N–65° N) (grey fill), aerosol concertation (A0.25) in Western Siberia at Fonovaya station (black line), the total number of hotspots (HS) (90° E–120° E, 55° N–65° N) (red line);

(2)    Middle graphs: the day-to-day variation of total column water (tcw, purple line) and surface wind speed (ws, green line) for the center of forest fire area 105° E, 60° N;

(3)    Upper graphs: the day-to-day variation of the total number of the lightning strikes (LS, yellow color), precipitation for the center of the forest fire area 105° E, 60° N (precip_north, light-blue color); precipitation: for June in point 54.5° N, 97.5° E, for July—the average amount for two grid points: 51.5° N, 103.5° E and 51.5° N, 104.5° E (precip_south, pink color).

Time-longitude cross-section in Figure 1b shows *GHGS* (black line) and surface temperature anomalies average for 55° N–65° N (red and blue fill). *GHGS* > 0 corresponds to blocking, *GHGS* > −2—near blocking. Light blue vertical line—high precipitation events. Yellow vertical line—start and finish of forest fire period.

Table S1 (in Supplementary Materials) is the collection of the chronology of precipitation (which has led to the floods), forest fires, blocks over Eastern Siberia, and RWB for Eastern Siberia and western border territory (60° E–120° E).

Figure 2 displays the forest fire clusters (FFCs), which were identified using the DBSCAN algorithm based on GFAS data from 1 June to 31 August. We combined the groups based on the time of FFCs existence in the following manner. Blue color—groups existed during 1–15 July, red—16–31 July. green—1–12 August.

Based on Figures 1 and 2 and Table S1 (Supplementary Materials), we divided the period from 1 June to 15 August into six time intervals: 1–19 June (preliminary period), 20–25 June (the first flood), 26 June–5 July (first forest fire large cluster formation), 6–21 July (second and third forest fire cluster formation and forest fire amplification), 22–28 July (the second flood), 29 July–15 August (forest fire period, amplification forest fires activity). In the analysis of the first period, which preceded the extreme precipitation and forest fires, we relied on published results regarding anomalies in Eurasia during June 2019.

*First (1–19 June)—Period preceding first blocking over the ES, extreme precipitation, and forest fires.* The most striking large-scale weather event over northern Eurasia in June 2019 was the record-breaking heat in Europe [50,70], https://climate.copernicus.eu/surface-air-temperature-june-2019, accessed on 11 March 2023.

We draw attention to the record-breaking high temperatures over Europe preceding the large-scale atmospheric events that we are studying over Siberia, which could possibly be associated with them. We do not exclude the possibility of such an association, relying on previous works describing teleconnection patterns over Northern Eurasia, such as AEA [29] and BBC [30], as well as previous works describing the record-breaking high temperature over Europe in June 2019 [50,70]. In these works, the record-breaking heat in Europe is analyzed in detail and associated with the AEA and BBC, respectively. Zhao et al. 2020 [50] based on NCEP–NCAR reanalysis data showed that a strong anomalous anticyclone appeared over Europe in June of 2019. The southerly wind anomalies to the west side of the anomalous anticyclone transported warmer air from lower latitudes towards Europe, contributing to the increase in surface air temperature there (Figure 2a in Zhao et al. 2020 [50]). The wave train with a barotropic vertical structure extended eastward from high latitudes in the North Atlantic across Europe to the Russian Far East (Figure 2b in Zhao et al. 2020 [50]). Positive anomalies in geopotential height were observed over Eastern Europe and Eastern Siberia, while negative anomalies were present over the Kara Sea East. (Figure 2b in Zhao et al. 2020 [50]). The authors have attributed this wave train to the AEA teleconnection pattern.

Xu et al., 2020 [70] using monthly mean Japanese 55-year reanalysis (JRA-55) demonstrated an anomalous anticyclone in the upper troposphere and associated it with the BBC

pattern. The authors demonstrate that the anomalous anticyclone results from an unusually intensified British–Baikal corridor (BBC) pattern and a synoptic Rossby wave breaking (RWB) event over Europe. The authors describe three sub-monthly heat waves during June 2019. The first two were associated with the BBC pattern, and the third was related to the combination of the previous BBC pattern activity and the RWB event, Figure 5 (in [70]).

It is worth noting that during the first half of June, the distribution of atmospheric blocking over Eurasia exhibited the "one by one" type of the "Europe + Russian Far East" pattern described in [28] (the longitude time cross-section of the *GHGS* blocking index for summer 2019 is in Supplementary Materials, Figure S1).

Figure 6 in Zhao et al. 2020 [50] shows that the June AEA index in 2019 was the most positive since 1979. A positive AEA index is characterized by the high geopotential height over the North Atlantic Ocean, Eastern Europe, Mongolia–north China, and low geopotential height over the northeastern North Atlantic Ocean and the Kara Sea–Northern Siberia. We have supposed that the extreme AEA teleconnection in June 2019 caused the pressure pattern of "the deep tropospheric trough in Western Siberia/high amplitude ridge over Eastern Siberia" and as a result, a strong baroclinic zone appeared between Western and Eastern Siberia Figure 2 in [29]. The AEA index remained positive up to the 20th June 2019 [50]. The wave activity flux (WAF) was pronounced in the North Atlantic-European sector from 1 June to 15 June (Figure 3, Interval 1).

*Second—Period of the first flood wave (20–25 June).* Up to 20 June, the circulation pattern over the Atlantic–Eurasia region corresponded to the positive AEA index, with a trough over the Kara Sea [50]. Between 20 and 21 June, a strengthening of WAF (Figure 3, Interval 2, −50 W–0 E) and deepening of the trough were observed (Video S1 in Supplementary Materials). On 21–22 June, a part of the trough was cut off and started to move towards the southeast (Video S1 in Supplementary Materials); on 24–25 June, a cutoff low was formed (Figure 4), with the center—53° N, 83° E. The weak rainfall in the front part of the cutoff low started on 23 June (Video S2 in Supplementary Materials), and on 24 June, the precipitation became extreme. The quasi-stationary rainfall belt was located over the Eastern Sayan Mountains during 24–26 June (Figure 5a, Video S2 in Supplementary Materials, Figure 6a,b). It caused an extreme flood in the Irkutsk oblast (Tulun, Nizhneudinsk, Shitkino) [10]. The cyclonic overturning of PV = 4–8 PVU was revealed for 25 June. Still, the first signs of PV contours deformation were observed on 24 June and maintained until 26 June. The overturning region is located near the border between the East Asia summer monsoon (EASM) and the polar vortex intrusion [71]. In the cyclonic part of the breaking, there were ascending vertical motions of 1.3 hPa/s (as shown in Figure 7a), while in the anticyclonic (warm) part, descending motions were observed. In the area located under the warm part of the breaking, water vapor fluxes were intensified (according to Era-Interim date, Figure not shown). The EASM jump was observed on 25 June, which is evident in Figure 5 and Video S2 based on the streamline pattern, and in Figure 8a,b based on the change of the total column water between 23 and 25 June.

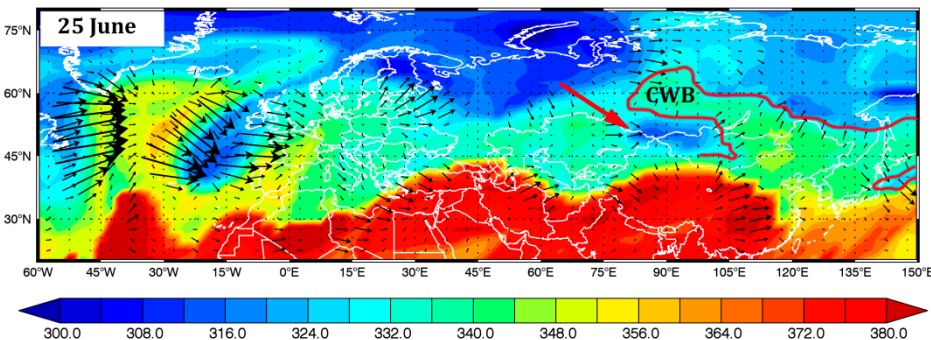

**Figure 4.** Circulation pattern (WAF—black arrows, m$^2$/s$^{-2}$ and PV-Θ—fill, K) on 25 June (2nd period). Here and below the red arrow show the position of cold PV-Θ advection toward Siberia, the red curve demonstrates the horizontal scheme of RWB. Era-Interim data.

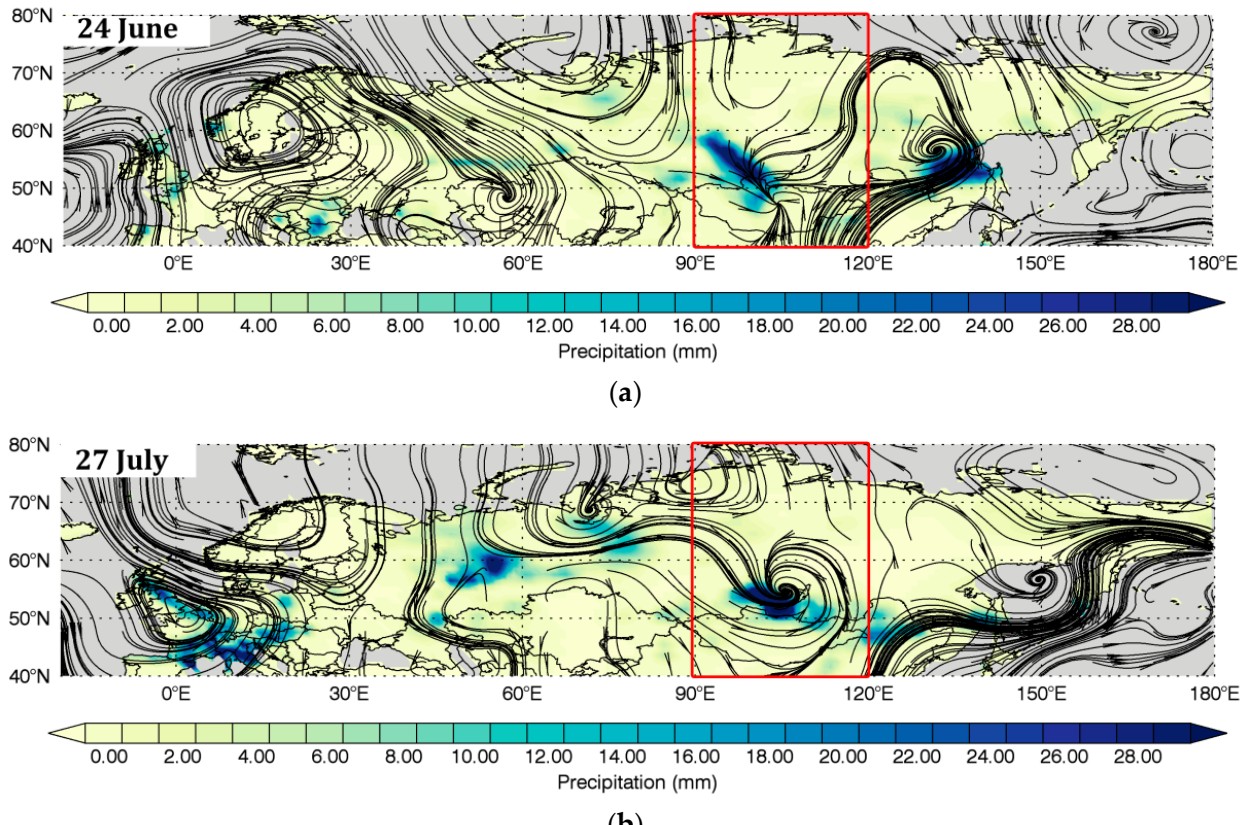

**Figure 5.** The total amount of precipitation and daily streamline for the first flood wave ((**a**) 24 June) and for the second flood wave ((**b**) 27 July). The red color—the region of interest. Era-Interim data, at 850 hPa.

So we have concluded that the CWB and precipitation were caused by the deep intrusion of polar air masses and were observed simultaneously. The extreme rainfall can be attributed to the wave breaking. We believe that two main factors contributed to the scenario of intrusion and breaking on 20–25 June 2019: an intense wave flux associated with an unusual positive AEA teleconnection throughout June and the characteristics of the Kara Sea trough during 20–23 June.

*Third—Period of first forest fire large cluster formation (26 June–5 July)* (Figure 9, Video S3 in Supplementary Materials). On 26 June (Video S3), the first blocking in westerly flow (*GHGS* > 0, Figure 1b) was detected over the longitudes of Lake Baikal. Furthermore, the intensification of the WAF from the North Atlantic continued, and it propagated towards the east (26–30 June). As a result of wave propagation into Siberia, two warm PV-Θ waves (28 June and 29 June–2 July) maintained the blocking with a vast anticyclone in the northern part of Siberia (the area depicted schematically based on the geopotential maps). The wave-like structure is evident from 26 June to 1 July (Figure 3, Interval 3, 0 E–50 E).

The first blocking over Eastern Siberia (Figure 1b) was caused by the intrusion of a low PV-Θ air mass, a cutoff low, and CWB, as well as the simultaneous intensification and propagation of WAF from the North Atlantic to Siberia. The subsequent AWB (Figure 9a) was a consequence of blocking and the propagation of WAF. The AWB also served as an additional source for the maintenance of blocking (the anticyclone in North Siberia, Video S3). The intense advection of heat from lower latitudes was associated with patterns on 28 June and 2 July (high PV-Θ). Clusters of fires began to form on 3 July and persisted until 15 July (Figure 2, blue color).

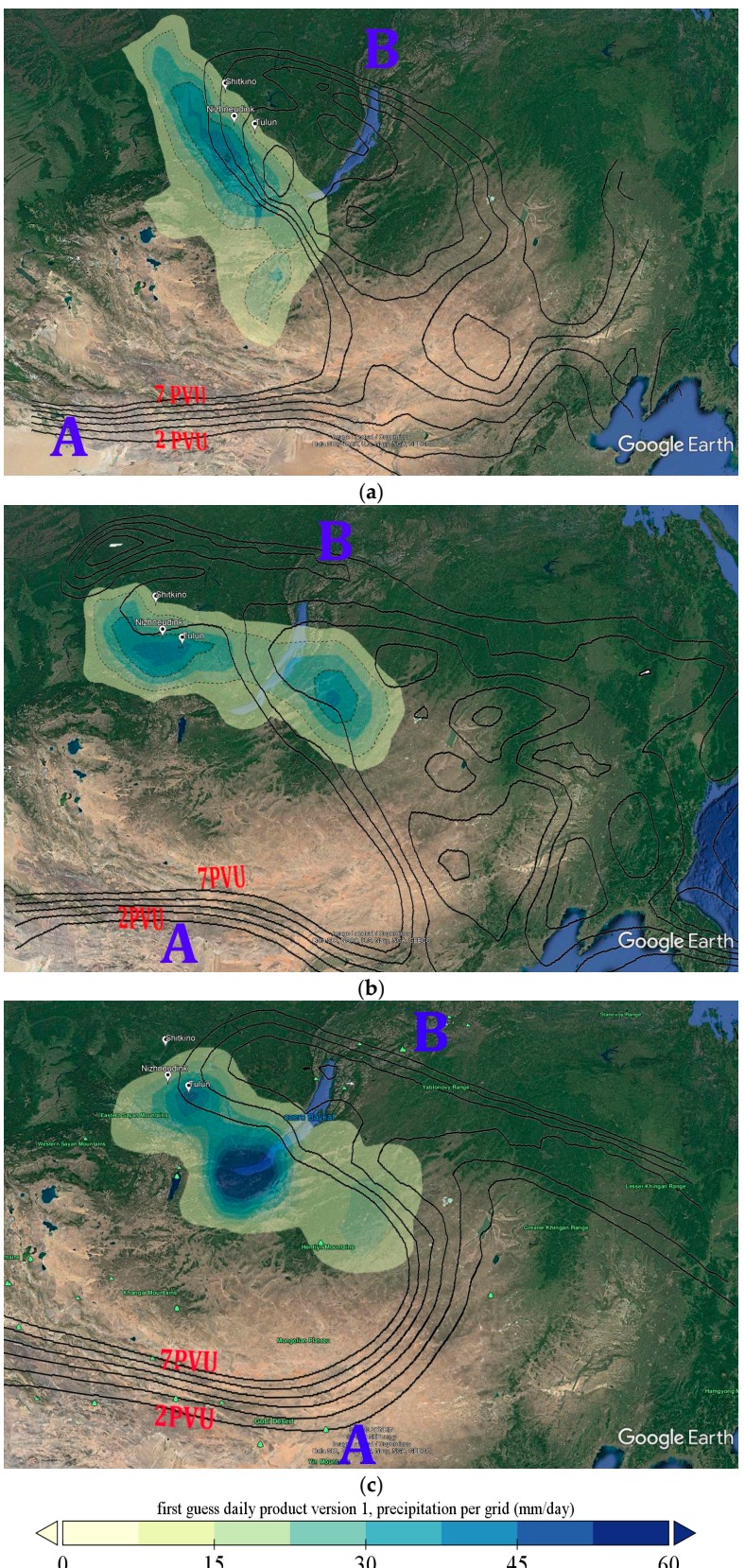

**Figure 6.** Goggle maps with the position of precipitation zone (total daily for 24 (**a**), 25 (**b**) June and 27 July (**c**)) and PV overturning (2–7PVU) for 24 June 18 UTC (**a**), 25 June 12 UTC (**b**) and 27 July 12 UTC (**c**), letters A and B shows the schematic position of the start and end cross-section in Figure 7. Era-Interim data.

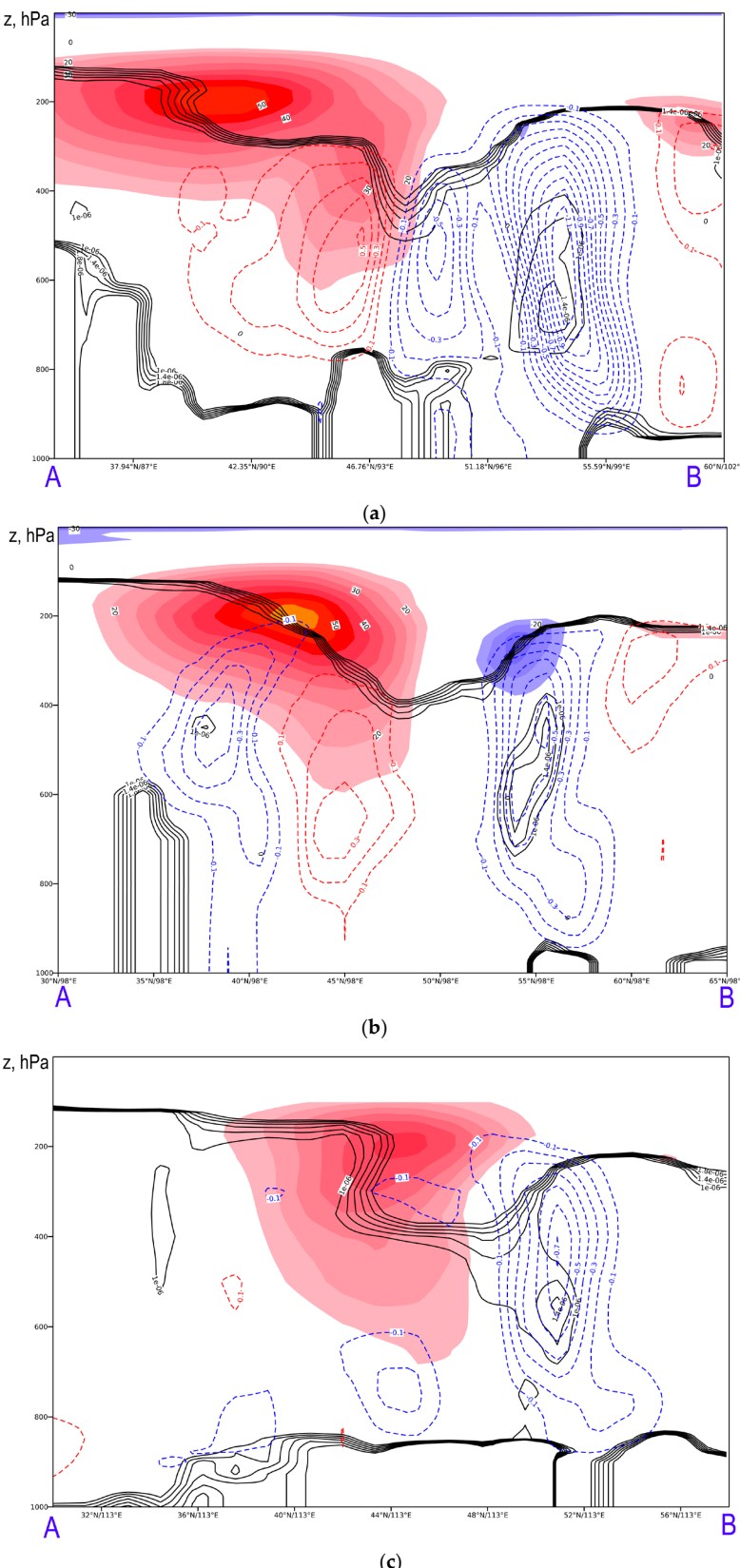

**Figure 7.** Cross-sections of horizontal wind speed (m/s) (blue and red fill), PV (PVU) (black lines) and vertical speed (Pa/s) (red and blue dashed lines) for 24 June 18 UTC (**a**), 25 June 12 UTC (**b**) and 27 July 12 UTC (**c**), letters A and B in Figure 6. The negative value for vertical velocity—ascending motions. Red shading—westerly wind (positive); blue—easterly wind (negative). Era-Interim data.

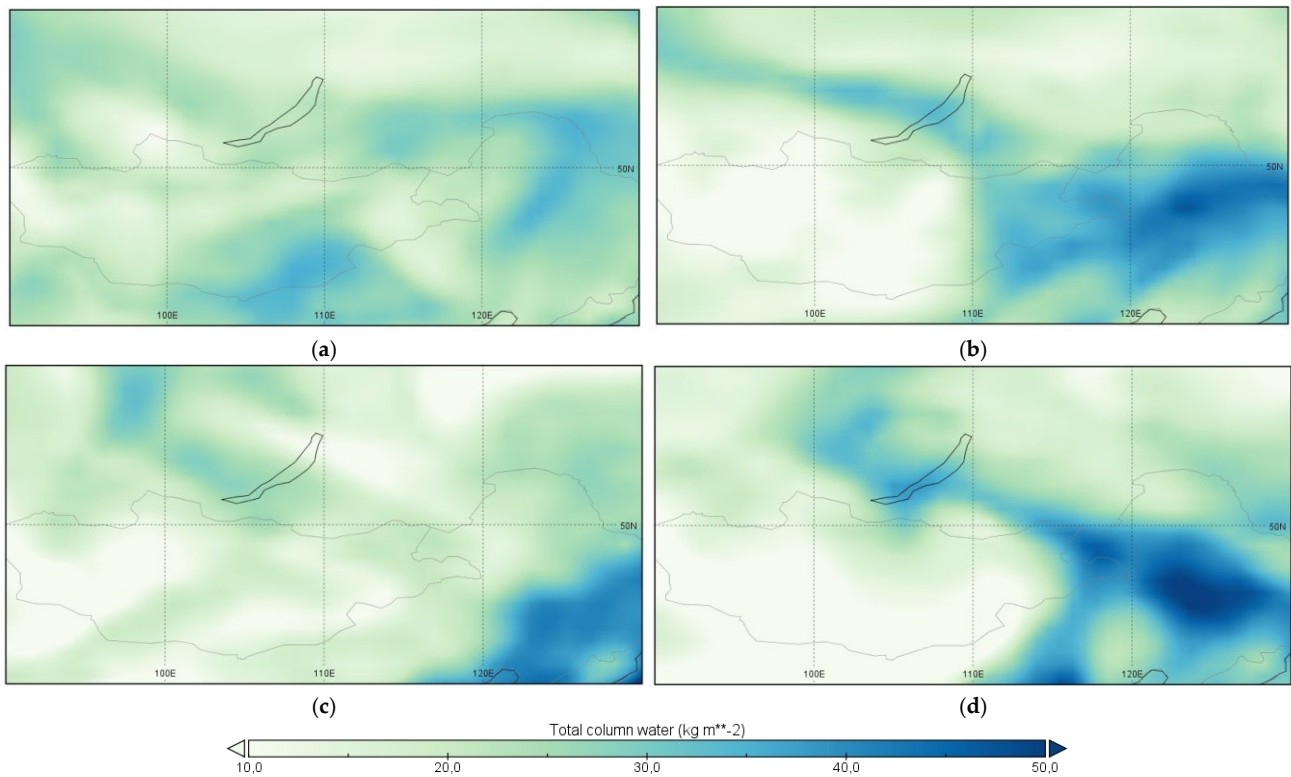

**Figure 8.** Total column water for maximal EASM jump during cyclonic wave breaking 23–25 June (**a**,**b**) and 26–27 (**c**,**d**) July 2019. Era-Interim data.

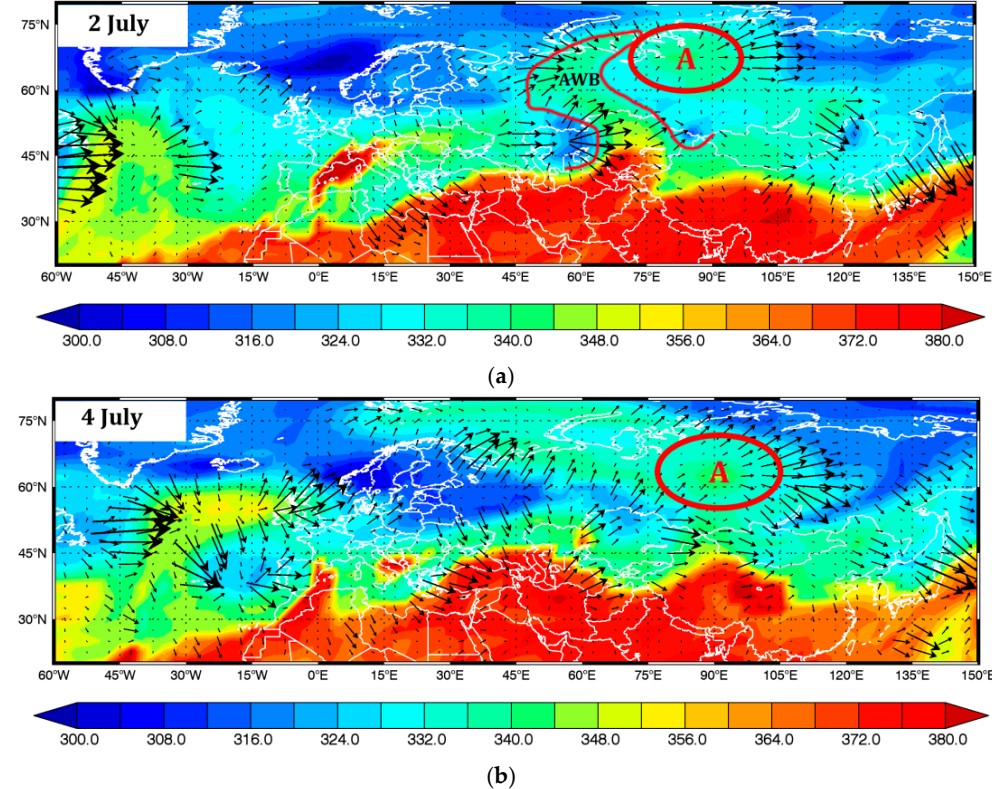

**Figure 9.** Circulation pattern (WAF and PV-Θ, similar to Figure 4) on 2 (**a**) and 4 (**b**) July (3rd period. Here and below, a red circle with the letter "A" marks the position of the blocking anticyclone (according to maps of geopotential at 500 hPa). Era-Interim data.

*Fourth—Period of second and third forest fire region formation and forest fire amplification (6–21 July)* (Figure 10, Video S4 in Supplementary Materials). On 5–6 July, the anticyclone in the northern part of eastern Siberia began to decay (4 July in Video S3, 6–8 July in Video S4). The transformation of the air masses associated with the previous period with the blocking anticyclone over ES can be traced (4 July in Video S3, 6–8 July in Video S4). The high PV-Θ air masses in the northern part of Eurasia have moved to North Europe. On 10 July, the trough that had been observed over North Europe since 4 July intensified. It was due to the amplification of WAF and the advection of cold air masses in the front part of the anticyclone. The WAF in the front part of the trough increased from 10–14 July; however, the synoptic waves do not propagate well into the northern part of ES (Figure 3, 10–30 July, Intervals 4 and 5). From 45° E to 150° E, the weak PV-Θ latitudinal gradient was observed (and weak geopotential height gradient Figure 1). For the period from 14 July to 17 July, two PV contours breaking were detected, first AWB (14–15 July) and second CWB (14–17 July) (Figure 10a), with a renewing cutoff low between them. Starting from 15 July, the large forest fire clusters in the second and third forest fire regions began to form (Figure 2, red clusters). The crucial role for forest fire spreading belongs to the CWB to the east of Lake Baikal (Figure 10a). Figure 11 shows the schematic maps based on the Worldview satellite image of hotspots for 16 and 17 July and potential vorticity levels. The increase in the number of hotspots and biomass emissions, as well as the northward shift of the forest fire area (as shown in Figure 2), occurred concurrently with the occurrence of CWB on 16–17 July. On 16 July, the anticyclone in the northern part of Siberia amplified. On 21 July, the anticyclone decayed simultaneously with the regeneration of the cutoff low (Figure 10b). Blocking over Siberia was observed until 22 July.

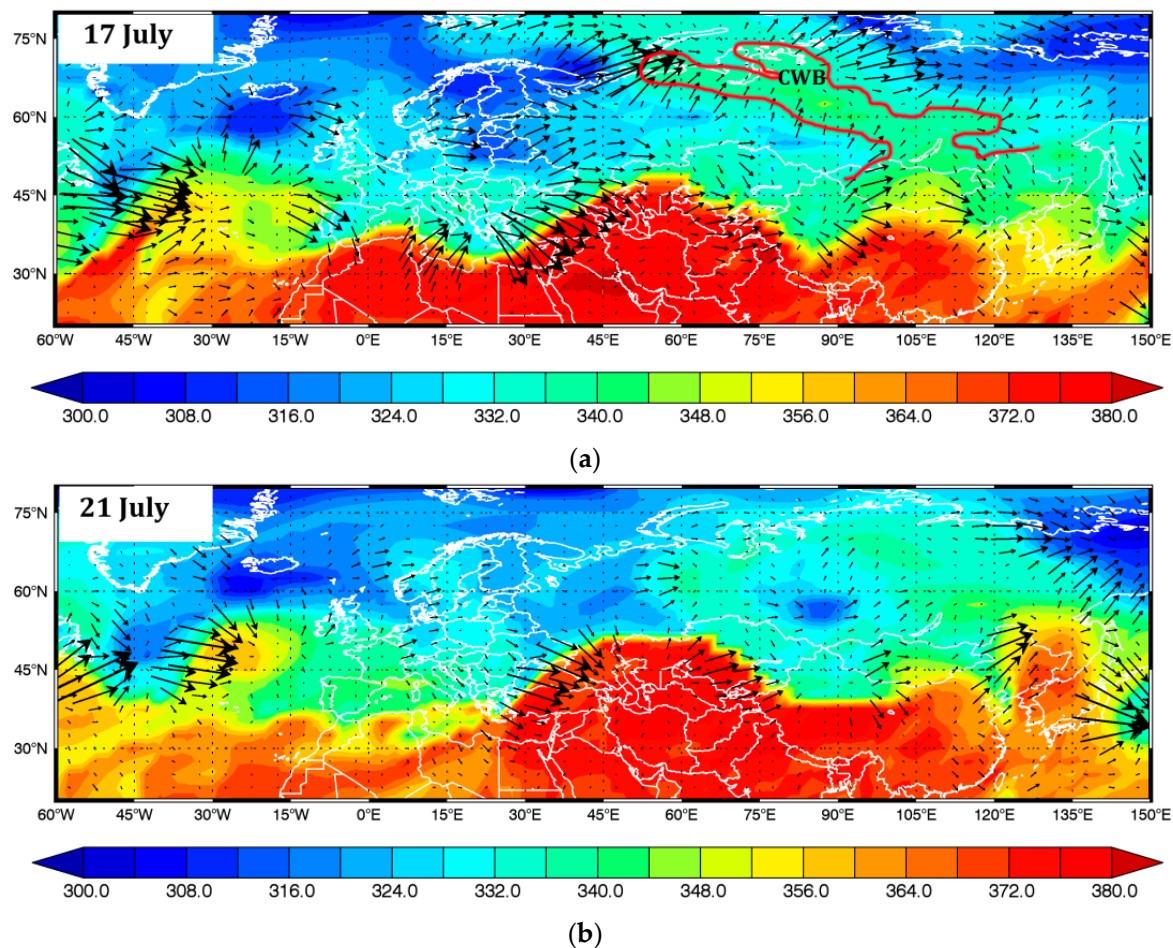

**Figure 10.** Circulation pattern (WAF and PV-Θ, K) on 17 (**a**) and 21 (**b**) July (4th period). Era-Interim data.

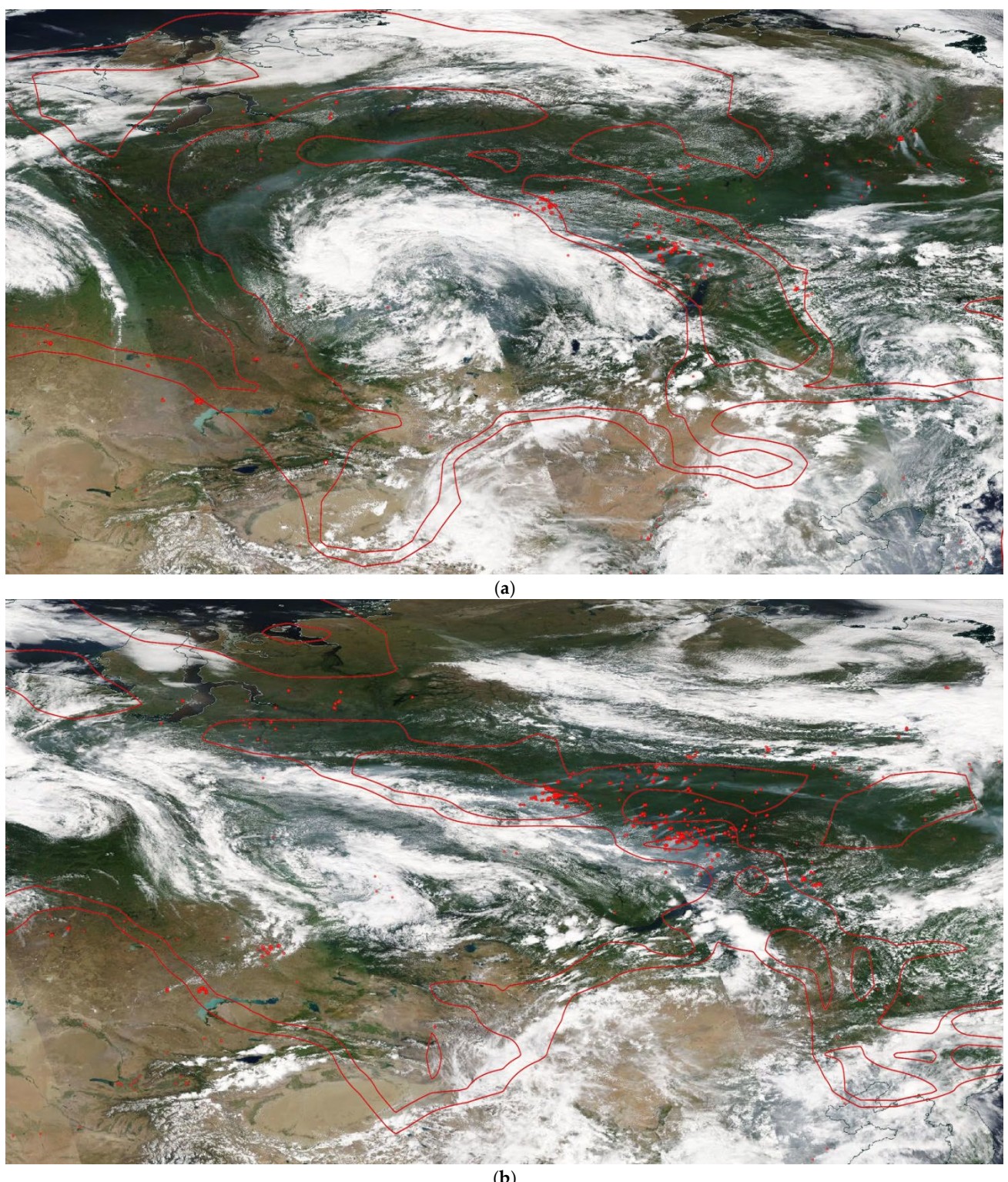

(**a**)

(**b**)

**Figure 11.** Worldwide maps with the hotspots position and schematic PV breaking (2 and 7PVU) according to Era-Interim on 16 July 12 UTC (**a**), 17 July 12 UTC (**b**), 60–120 E, 30–70 N, Era-Interim data.

*Fifth—Period of the second flood formation (22–28 July)* (Figure 12). On 22–28 July, two main large-scale dynamical events can be identified (Video S5 in Supplementary Materials). First, it is anticyclonic breakings over Europe and increased WAF associated with them (Figure 3, Interval 5), (Figure 12, Video S5). Second, it is the transformation and eastward movement of the low PV-Θ area over sector 60° E–90° E (Figure 12, Video S5), including the

reinforced cutoff low on 21 July (Video S4, 90° E). During 21–28 July, the low PV-Θ moved to the east along the area of PV-Θ high gradient (45° N, 90° E–120° E). On 23, 27, 28 July the CWB was detected over 45° N–60° N, 90° E–120° E with maximal reversing on 27–28 July (Figure 6c). The main features of CWB 26–28 July were similar to the CWB 25–26 June: the increase in vertical motions (up to 0.7 hPa/s) (Figure 7c), the sharp drop of tropopause height between the south and the north part of breaking (not illustrated here) and high column water in the atmosphere (Figure 8c,d). The high column water can be related to the increase and northward jump of the EASM in the lower troposphere between 26–27 July (Video S2, Figure 8). The maximum intensity of rainfall (Video S2) in the southern part of eastern Siberia was observed during the period of maximum gradient of overturning 27–28 July (Figure 6c). The CWB in Eastern Siberia coincided with the occurrence of the AWB over Europe (as shown in Figure 12 and Video S5) and an increase in WAF (as shown in Figure 3, Interval 5). Regarding the formation of blocking, the CWBs on 26–28 July are similar to the CWB that accompanied the first precipitation period (25–26 June). Although the geopotential field at the end of July did not indicate the presence of a blocking pattern (*GHGS* < 0) (Figure 1), the overturning in the region helped maintain a low PV gradient over Eastern Siberia.

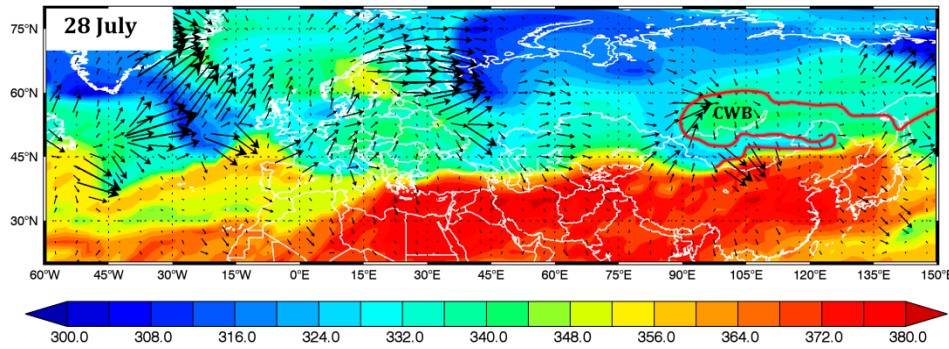

**Figure 12.** Circulation pattern (WAF and PV-Θ) on 28 July (5th period).

Sixth—The last forest fire period, secondary amplification forest fires activity in three regions, end of the blocking period (29 July–15 August) (Figure 13, Video S6 in Supplementary Materials) The pattern is similar to 26 June–2 July (third period); after the CWB two breaking from the west, leading to the strengthening of ES-blocking high (Figure 13, Video S6). Two AWBs (2 and 7 August) at 75° E–90° E occurred due to WAF increasing in the front part of the trough formed after breaking over Europe during 26–31 July (Videos S5 and S6). Figure 1b shows that the blocking had fluctuations during 1–12 August, with the peak of *GHGS* on 3–4 and 8 August. On 3–8 August, the number of forest fire hotspots and biomass burning emissions increased (green forest fire clusters, Figure 2) however, in most parts, for hotspot number than for emissions (Figure 1a). After 12 August, the trough was finally established over Eastern Siberia. The general WAF over Eurasia moved to subtropics (12 August in Video S6).

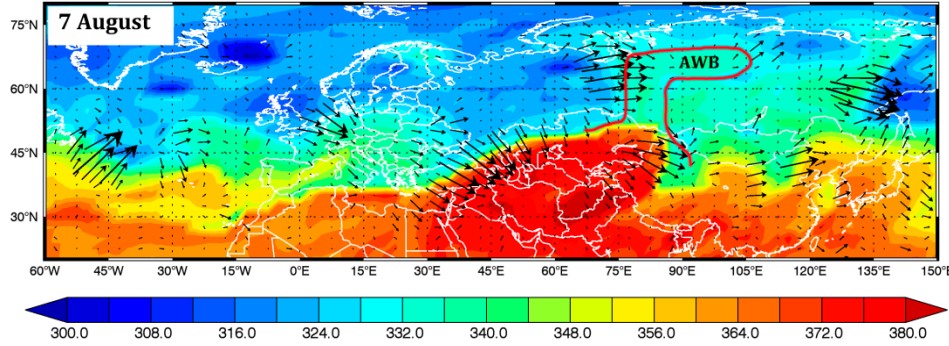

**Figure 13.** Circulation pattern (WAF and PV-Θ) on 7 August (6th period). Era-Interim data.

### 3.2. The Factors Contributing to Forest Fire Ignitions

Anthropogenic sources have been identified as the significant cause of Russian wildfires [5,72]. However, in 2019 a mega wildfire in Siberian was observed in hard-to-reach areas of Siberia (in northern Krasnoyarsk Krai, the Sakha Republic, and Zabaykalsky Krai) and was reported as being caused by natural factors of ignition. According to a quote from the Krasnoyarsk Forest Fire Center by the Russian News Agency TASS, the causes of forest fires were natural and were due to a 30-degree Celsius heat (86 degrees Fahrenheit), gusts of wind, and dry thunderstorms. (https://www.nasa.gov/image-feature/goddard/2019/huge-wildfires-in-russias-siberian-province-continue, accessed on 11 March 2023).

In Figure 1a, the yellow bar chart shows the total amount of lightning strokes (LS) in the forest fire area. The LS increase occurred between the first day of the first blocking (Table S1 in Supplementary Materials, 27 June) and the last day of the second blocking (19 July). Figure 14a,b shows anomalies in LS for the 1–10 and 10–20 July along forest fire clusters obtained for these periods. Figure 14c–e shows the days with the highest LS. In July 2019, LS levels higher than those in the previous period of 2009–2018 were observed; furthermore, in most of Eastern Siberia, the LS were not accompanied by rain higher than 3 mm (Figure 1, blue bar chart, Figure 14c–e).

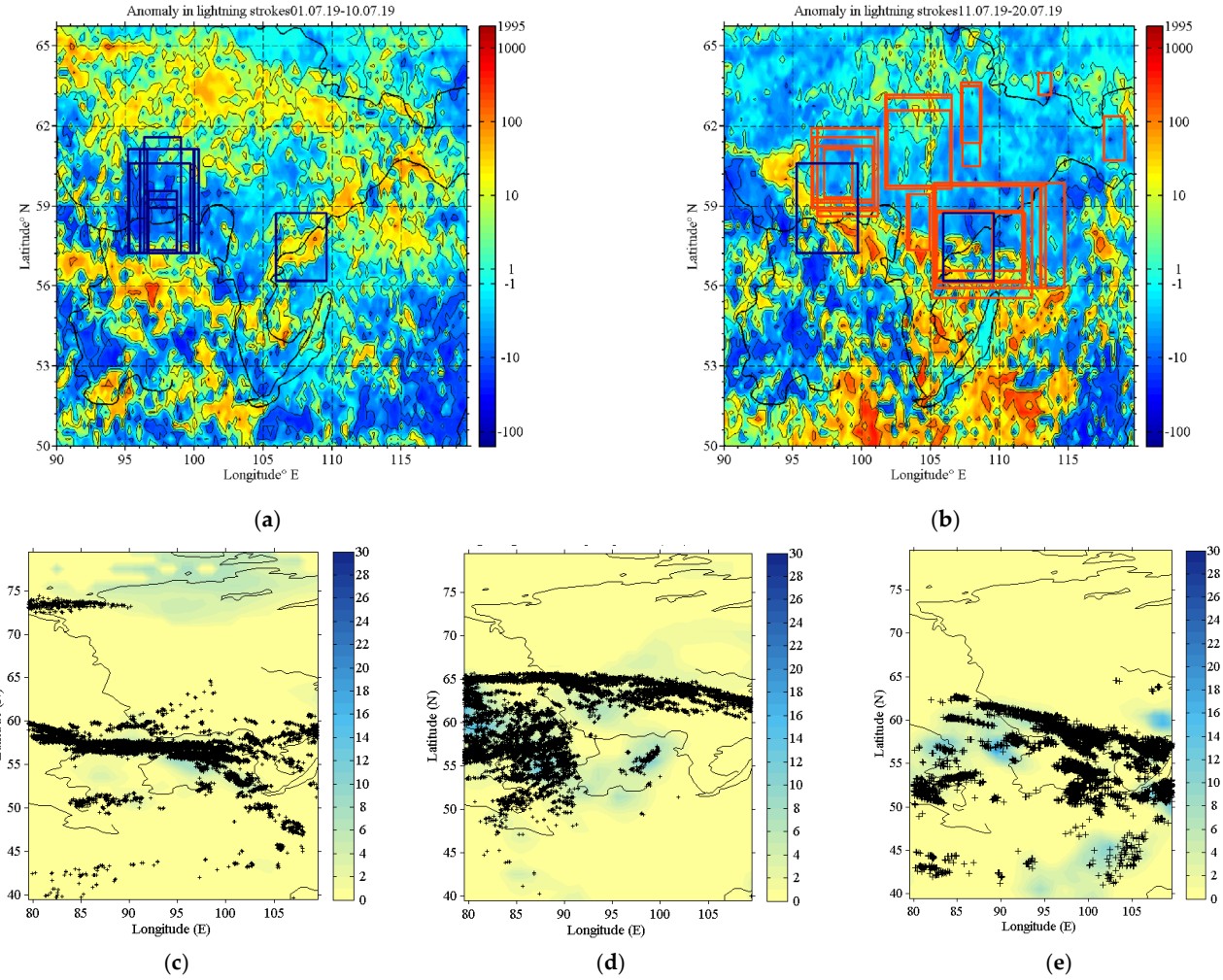

**Figure 14.** Anomaly in lightning strokes compares to periods 2009–2018 (**a**);1–10 July 2019 (**b**);10–20 of July 2019. The total amount of lightning strokes and precipitation on 2 (**c**), 6 (**d**), 17 (**e**) July.

Thunderstorms that occur without significant rainfall on the ground are called "dry thunderstorms" (DTh) [73,74]. DTh are the most dangerous in the case of dry soil and atmospheric conditions. More importantly, dry thunderstorm events often lead to large

wildfire outbreaks [73]. The increase in DTh during blocking can be explained by the increase in the vertical instability of the atmosphere. The instability is associated with the formation of cutoff filaments with a low potential temperature (or high PV) [73]. The example on 17 July shows that the increase in DTh during blocking can be explained by the vertical instability of the atmosphere (Figure 10a) and high near-surface temperature (Figure S2b). In Figure S2a, CAPE (convective available potential energy) is shown, which characterizes the amount of energy available for convection in the atmosphere. The combination of instability and high surface temperatures is due to the configuration of the cyclonic breaking on 17 July.

The estimates of the effect of LS on fires and the ratio of their contribution require additional research. In the present paper, we only draw attention to the increase in the number of dry thunderstorms during the 2019 blocking periods. Additionally, our findings are in agreement with the statement made by the Krasnoyarsk Forest Fire Center.

*3.3. Factors That Affected the Magnitude of Forest Fire Intensity (Hotspot Number and Emission)*

We have shown that forest fire periods in 2019 are strongly associated with wave breaking and atmospheric blocking. Both breaking and blocking caused the positive surface temperature anomalies in Northern Siberia (Figure 1b). There are three main effects of surface temperature increase linked to breaking and blocking anticyclone formation. The first effect is the transport of air masses from lower latitudes [75,76], which is indicated by high PV-Θ values on the maps. The second effect is the adiabatic warming in the upper-level anticyclone, caused by the subsidence of air parcels [77]. Additionally, the downward motion anomalies associated with blocking anticyclones result in less cloud cover and increased incoming solar radiation. The latter also contributes to an increase in surface temperature [50]. Ponomarev et al. (2016) [3] demonstrated the significant relationship between forest fire characteristics in Siberia, surface temperature, and incoming solar radiation. Thus, regardless of the contribution of the three factors, we evaluated the role of wave breaking and blocking as crucial drivers of surface temperature anomalies in summer 2019.

However, it is necessary to investigate the other effects of blocking in Siberia, which contribute to the maximum magnitude of CO emissions and hotspot count (Figure 1a): atmospheric moisture and wind speed [75,76]. Therefore, we analyzed the variations in the total atmospheric moisture content and wind speed during July-August (two critical factors, along with temperature and precipitation, included in fire hazard calculations) for three blocking events (Table S1 in Supplementary Materials) (Figure 1b, tcw and WS). We tracked the cloud cover based on Worldview maps (https://worldview.earthdata.nasa.gov/, accessed on 21 December 2021) along with moisture and wind speed. Video S3 (4 July), Video S4 (19 July), and Video S6 (4 August, 8 August) illustrate the circulation patterns that correspond to the blocking life cycle from the last RWB to blocks decay, as shown in Table S1. During these periods, there is an amplification of WAF and intrusions of cold air masses downstream of the blocking anticyclone in North Siberia. The location of the blocking anticyclone is the most favorable for forest fire spreading. At the same time, the forest fire area is located in a region with minimal cloud cover (https://go.nasa.gov/2NHy6Ik, accessed on 21 December 2021, https://go.nasa.gov/3idmNpt, https://go.nasa.gov/2ZknNzj, accessed on 21 December 2021, https://go.nasa.gov/2YLBzMp, accessed on 21 December 2021) and near the downstream area under the intrusion of cold air masses. For all periods, a decrease in total column water was observed (Figure 1a). The significant reduction in total column water in the lower atmosphere, up to 10 kg/m$^2$, led to amplified fire danger. On June 20th, there was a cold air intrusion from the east (Figure S2c), which resulted in an increase in geopotential contrasts (Figure S2d). The zone of low moisture content in the north expanded (Figure S2e), and the near-surface wind intensified (Figure S2f). This period is considered critical and, combined with the increase in lightning activity on 17th July, may have contributed to the maximum increase in fires.

The deepest low PV-Θ intrusion in the front of the anticyclone was observed in mid-July (17–19 July, Video S4). Mid-July is the period of maximum intensification of CO emissions and the number of hotspots (Figure 1a). Additionally, Figure 1a shows that in mid-July the fire area experienced a maximum decrease in total column water and a maximum increase in surface wind speed. The advection of cold PV-Θ resulted in the renewal of the cutoff cyclone in the south of Western Siberia on 21 July (Figure 10b). The direction of cold advection was extremely atypical for the summer period. Figure 15 displays the geopotential height at 500 hPa and the total amount of hotspots for 12–23 July 2019. The massive block was located over Siberia, and the cutoff cyclone with a center of 55° N–90° E promoted the transfer of smoke plumes from fires to the west of Siberia. In mid-summer, as shown in Figure 1a, the air composition in Western Siberia, which is not subject to fires, changed simultaneously with the change in fire intensity. The area from the Ural Mountains to the Far East located under a blocking dome was filled with smoke from fires (https://go.nasa.gov/2CBktIx, accessed on 21 December 2021).

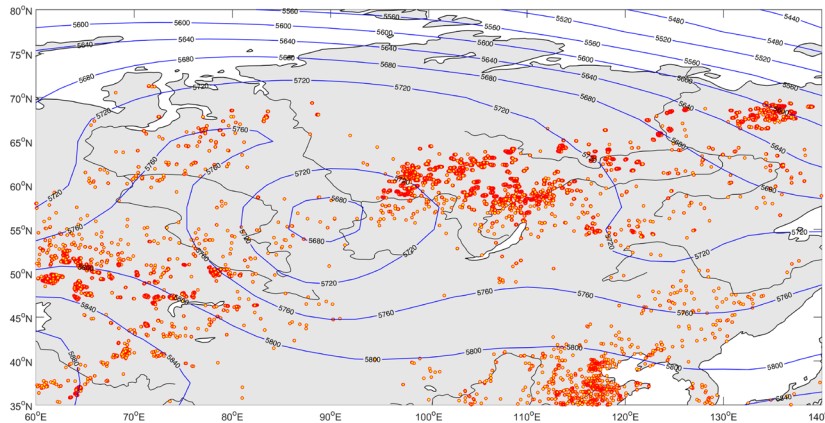

**Figure 15.** The total number of hotspots (FIRMS) and geopotential height at 500 hPa (12–23 July 2019) (Era-Interim).

### 3.4. The Factors That Affected the Extreme Precipitation and Flood

As forest fires floods in 2019 are strongly associated with waves breaking and atmospheric blocking (Figure 1b). The main cause of floods is precipitation [4]. Precipitation intensity depends on ascent condition (instability) and total column water of air mass. The three main mechanisms of air instability generation are local heating, orography (movement of air up the slopes of mountains) and large-scale potential vorticity dynamics (strengthening of meridionality).

We have shown that in 2019, extreme precipitation over the southern part of Eastern Siberia was due to a large-scale strengthening of meridionality: increased wave activity over Europe (Figure 3, periods 2 and 5) and cyclonic type of Rossby wave breaking over Eastern Siberia. This large-scale mechanism of instability was strengthened by the orographic mechanism; atmospheric front, precipitation area and ascending movements were localized clearly along the Eastern Sayan ridge (Figures 6 and 7). The contrast between warm moist air from the East Asian summer monsoon area (Video S2) and cold arctic air from the Kara Sea (Videos S1 and S5), which was involved in the southern part of Eastern Siberia during the cyclonic Rossby wave breaking, led to an increase in the strength of the vertical circulation associated with frontogenesis. This resulted in extreme precipitation, contributing to the 2019 Siberian summer anomaly.

## 4. Discussion and Conclusions

Record-breaking forest fires and floods were observed in Eastern Siberia (ES) between 24 June and 12 August in 2019. We investigated these events as one phenomenon due to the synoptic (Rossby) waves breaking (RWB) and atmospheric blocking (blocks). We

have demonstrated periods for RWB based on overturning potential vorticity (PV) contour; blocks based on geopotential gradient; transport and transformation of air masses based on potential temperature on the dynamical tropopause (PV-Θ); forest fire based on carbon monoxide emission and hotspots; flood based on the total amount of precipitation. Every specific period was described by synoptic analysis. We have obtained the following key results:

1.  The rainfall and forest fire in Siberia in 2019 were strongly associated with wave breaking and the life-cycle of blocking-high in the northern part of Siberia. The blocks were formed and maintained by two types of RWB: cyclonic type (CWB) from the east ES (110° E–115° E) and anticyclonic type (AWB) to the west ES (70° E–90° E). The CWB that occurred in latitude belt 40° N–60° N did not lead to the formation of the blocking high in geopotential; nevertheless, the CWB caused a low gradient in PV around Lake Baikal (Eastern Siberia). The AWB and CWB that extended above 60° N resulted in the blocking high on geopotential in the northern part of ES. Both types of breaking occurred in the front part of the cutoff low or trough. The main CWB and AWB were observed: 24–26 June (CWB), 28–29 June/1–2 July (AWB), 14–17 July (both AWB and CWB), 25–28 July (CWB), and 2 and 7 August (AWB). According to the geopotential gradient, the blocking over ES was observed three times: 26 June–3 July, 12–21 July, and 4–10 August (with the break 5–7 August);

2.  The rainfall in the southern part of ES (24–25 June, 25–28 July) was associated with the baroclinic growth of synoptic eddies accompanying CWB around Lake Baikal. Depending on the degree of PV overturning, the rainfall can be quasi-stationary for some days. The total precipitation caused by breaking depends on the initial baroclinicity (vertical velocity) and the border East Asian summer monsoon (EASM). The EASM can sharply turn to the southwest and add extra precipitable water. CWB associated with extreme precipitation either preceded blocking or occurred after blocking decay (simultaneously with the eastward movement of the low PV-Θ part of the blocking). Additionally, CWB occurred with high precipitation and had lower amplitude (only up to 60° N) compared to AWB (occurring above 60° N);

3.  The periods of forest fires are associated with the establishment of blocking high in the northern part of ES, mainly due to high amplitude breaking from west of ES (75° E–90° E). The peak of forest fires was on 4–5 July, 19–24 July, 5–6 and 8 August; occurred in periods of blocking decay. The location of blocking anticyclones can be favorable for the spread of forest fires. The forest fire area is located simultaneously in areas with minimum cloud cover and near downstream areas where cold air masses are intruding. For all forest fire peaks, a decrease in water content was observed. The most extreme peak in forest fire intensity was related to an anticyclone resulting from double breaking from the west and east (Video S4). The periods of blocking decay were characterized by the intrusion of cold air masses along the eastern part of the blocking high. In addition to the decreasing total column water and cloud cover, an increase in surface wind speed was observed. These processes drove forest fire intensification and spreading. Furthermore, from 26 June to 19 July, the formation and decay of blocking were accompanied by dry thunderstorms (DTs). DTs occurred along the periphery of low PV-Θ filaments. DTs are potentially the primary cause of for fire ignition in Siberia regions with the lowest population density (above 60° N) [78];

4.  We have concluded that both types of extremes, namely forest fires in Northern Siberia and floods in Southern Siberia, are closely related. We demonstrated the relationship by synoptic analysis of wave breaking and blocking formation. Both types of wave breaking have been detected by PV overturning on 350 K, indicating that they can be associated with exchange related to the subtropical tropopause. The CWB occurred in the southern part of Siberia (45° N–60° N), caused extreme rainfall, and maintained a low PV gradient eastward of Lake Baikal (24–25 June, 22–28 July). The low PV (PV-Θ) gradient and strengthening of the wave activity flux from the Europe–Atlantic sector may be the reason why the AWB has a high amplitude westward towards the region

with low PV (28 June–2 July, 14–15 July, and 2–7 August). In turn, AWB creates the condition for the formation of blocking highs and strengthens the WAF in the northern part of Siberia, which affects temperature, cloud cover, wind speed, and moisture content. The complicated combinations of the CWB and AWB were the main drivers of the extreme forest fires. In the specific case of the summer of 2019, the repeated position of the blocking anticyclone three times before its decay played a crucial role.

We want to discuss the relationship between the variability of fire intensity in the cluster region, blocking, and wave breaking as a debatable issue. Figure 16 presents the total emissions of CO (from biomass burning) from 2003 to 2019, with the highest emissions recorded in 2019. This confirms the earlier findings of Ponomarev et al. (2016) regarding changes in wildfire numbers and burned areas in Siberia [3]. As stated in [3], the number of forest fires and the size of burned areas have increased (1996–2015). As highlighted by previous studies [79,80], rising temperatures and drier conditions have led to longer and more severe fire seasons, resulting in a significant increase in forest fires across North America and Siberia. Our paper further emphasizes the crucial role of atmospheric blocking in driving positive temperature anomalies, and underscores the need to monitor changes in Rossby wave breaking and blocking formation over Siberia in the future. It is not only important to track changes in their frequency, but also in the characteristics of RWB and blocking formation.

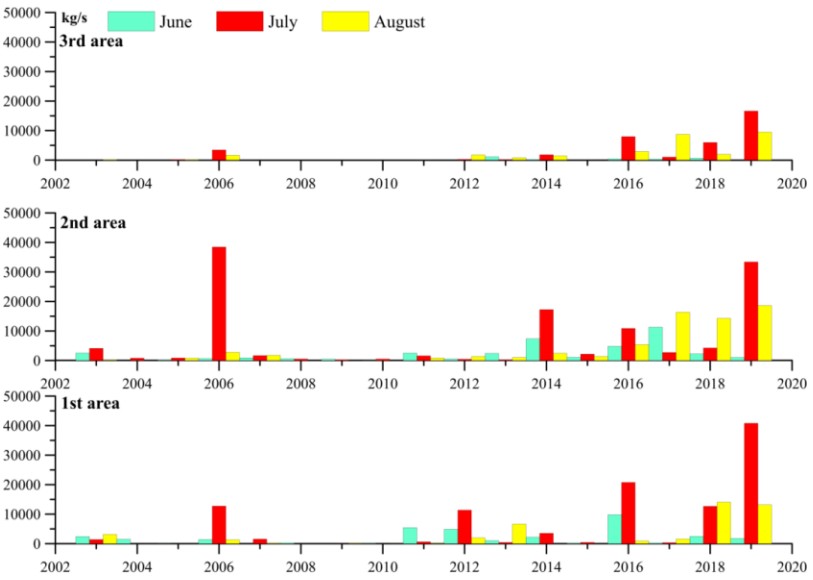

**Figure 16.** Total CO emissions (kg/s) from biomass burning in the cluster area are shown in Figure 2. Area 1st—58° N–63° N, 95° E–102° E, 2nd—56° N–61° N, 104° E–115° E, 3rd—61° N–64° N, 102° E–108° E (according Figure 2).

The development of the 2019 summer scenario, in our opinion, can be attributed to three key factors: the anomaly AEA index during June [50], the specific topography of the Kara Sea trough, and the increasing wave-train from the Atlantic during 21–26 June.

It is important to note that some researchers have suggested that there has been a change in the pattern of Rossby wave propagation over Eurasia since the mid-1990s [13,53]. Li and Ruan [29] have highlighted the increasing trend of the Atlantic–Eurasian teleconnection index over the past three decades. The evolution and amplification of the Rossby wave pattern can lead to modifications in the relationship between breaking and blocking. Moreover, the role of the positive feedback loop of "heat wave-soil moisture" may also increase [46,49,81,82]. The aridification of Eurasia in the 21st century [83] can further contribute to the change in the relationship between RWB, blocking, and wildfires.

**Supplementary Materials:** The following supporting information can be downloaded at: https://www.mdpi.com/article/10.3390/fire6030122/s1, Figure S1: The longitude time cross-section of the GHGS blocking index for summer 2019; Figure S2: 17 July 12 UTC (a) Convective amiable potential energy (CAPE), (b) surface temperature; 20 July 12 UTC (c) PV-Θ, (d) geopotential height at 500 hPa, (e) total column water, (f) wind at 10 m; Table S1: The time table of high precipitation, forest fires clusters formation, RWB and blocks; Video S1: Between 20 and 21 June, a strengthening of WAF (Figure 3, Interval 2, –50 W–0 E) and deepening of the trough were observed; On 21–22 June, a part of the trough was cut off and started to move towards the southeast; Video S2: The weak rainfall in the front part of the cutoff low started on 23 June, and on 24 June, the precipitation became extreme; Video S3: Period of first forest fire large cluster formation (26 June–5 July); Video S4: Period of second and third forest fire region formation and forest fire amplification (6–21 July); Video S5: Period of the second flood formation (22–28 July); Video S6: The last forest fire period, secondary amplification forest fires activity in three regions, end of the blocking period (29 July–15 August).

**Author Contributions:** Conceptualization, O.Y.A.; methodology, O.Y.A., A.V.G., P.N.A. and B.D.B.; software, L.D.T., K.N.P., P.N.A., Y.V.M. and A.V.G.; validation, L.D.T., K.N.P., P.N.A., Y.V.M. and A.V.G.; investigation, O.Y.A.; writing—original draft preparation, O.Y.A. and E.V.D.; writing—review and editing, O.Y.A., E.V.D. and P.N.A.; visualization, L.D.T., K.N.P., P.N.A., Y.V.M. and A.V.G.; supervision, B.D.B.; project administration, E.V.D. All authors have read and agreed to the published version of the manuscript.

**Funding:** This research was funded by the Russian Science Foundation № 23-27-00167 "Relationship between the formation of extreme precipitation in Southern Siberia and Rossby waves breaking and atmospheric blocking" (https://rscf.ru/en/project/23-27-00167/, accessed on 10 March 2023).

**Institutional Review Board Statement:** Not applicable.

**Informed Consent Statement:** Not applicable.

**Data Availability Statement:** Global Fire Assimilation System (GFAS) (CAMS Global Fire Assimilation System: http://apps.ecmwf.int/datasets/data/cams-gfas/, accessed on 12 December 2021). Fire Information for Resource Management System (FIRMS) (https://modaps.modaps.eosdis.nasa.gov/services/about/products/c6-nrt/MOD14.html, accessed on 12 December 2021) Hotspots visualizer Worldview based on the satellite image (http://worldview.earthdata.nasa.gov, accessed on 12 December 2021). Fonovaya station (http://lop.iao.ru/EN/fon/diffbat/, accessed on 12 December 2021) Worldwide lightning location network, WWLLN, http://wwlln.net/, accessed on 12 December 2021 Global Precipitation Climatology Centre (Deutscher Wetterdienst) (GPCC) (https://opendata.dwd.de/climate_environment/GPCC/html/gpcc_firstguess_daily_doi_download.html, accessed on 12 December 2021) ECMWF ERA-Interim reanalysis datasets (https://apps.ecmwf.int/datasets/data/interim-full-daily/levtype=sfc/, accessed on 12 December 2021).

**Acknowledgments:** The authors are very grateful to three anonymous reviewers who helped improve the article.

**Conflicts of Interest:** The authors declare no conflict of interest.

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
