# Peer review of "Effects of Rossby Waves Breaking and Atmospheric Blocking Formation on the Extreme Forest Fire and Floods in Eastern Siberia 2019"

_fire, doi:10.3390/fire6030122_

Round 1

Reviewer 1 Report

In 2019, the record-breaking floods and forest fires were observed in Eastern Siberia from June 24 to August 12. The authors investigated these events as one phenomenon, which occurred due to the synoptic (Rossby) waves breaking and atmospheric blocking. They revealed periods for Rossby wave breaking based on overturning potential vorticity contour; blockings based on geopotential gradient; and transport and transformation of air masses based on potential temperature on the dynamical tropopause.  Thereafter, they linked to (associated with) these atmospheric phenomena forest fire based on carbon monoxide emission and hotspots and flood based on the total amount of precipitation. Every specific period with extreme events of two types was described by synoptic analysis that supported their assessment.

 The authors concluded that the periods of forest fires are associated with the establishing of blocking high in the northern part of East Siberia mainly due to high amplitude breaking from west of it (in the 75°E - 90°E sector). It was found that the most extreme forest fire intensity peak was related to anticyclonic conditions resulting from double breaking from west and east.

 Finally, the authors concluded that in 2019, both types of extremes: the Northern Siberia Forest fire and Southern Siberia flood events are somewhat related. They demonstrated this relationship by synoptic analysis of wave breaking and blocking formation.

I have two major concerns:

·         The analyses and associations are made using the synoptic and forest fire information for one unusual summer season. How typical are these associations? Were they the same during the past forest fire seasons and/or will they work during the future years?  The authors have the data since 1979. Please, hint how typical were the relationships/associations that you describe.   I do not insist that the authors conducted a tremendous volume of analyses related to the entire period of instrumental observations and reanalyses. However, the words of caution in the Introduction and Discussion Sections of the manuscript about the sample size which, as the authors claim, was “one phenomenon” are warranted.  

·         High summer rainfall and floods in the continental extratropics are frequently related to regional convective storms.  Additionally, in Siberia, early summer floods are mostly related to the snowmelt (freshet).  This high flow period lasts until June. To link these high flow events to the large-scale atmospheric Rossby waves is an oversimplification. Or, at least, require a thorough justification.

Minor comments:

Line 176:  PVU - please, decipher the abbreviation (potential vorticity ??).

Table 1 column 5.  Please, decipher the meaning of footnote 1 in … (110E)1

Line 284. The AEA index for June was most positive in 2019 from 1979.   Please, edit English in this sentence. Possible variant could be:  The June AEA index in 2019 was the most positive since 1979.

Line 343. … fig.6 instead of fig. fig. 6.

Line 366. … framework … (edit, please).

Line 503.  Please, prove this suggestion.

Author Response

We are grateful to the Reviewer for the careful reading of the article and for useful comments. According to the comments, we have made the following changes to the text of the article:

Major concerns:

  •         The analyses and associations are made using the synoptic and forest fire information for one unusual summer season. How typical are these associations? Were they the same during the past forest fire seasons and/or will they work during the future years?  The authors have the data since 1979. Please, hint how typical were the relationships/associations that you describe.   I do not insist that the authors conducted a tremendous volume of analyses related to the entire period of instrumental observations and reanalyses. However, the words of caution in the Introduction and Discussion Sections of the manuscript about the sample size which, as the authors claim, was “one phenomenon” are warranted.  

Answer:   Thanks a lot for the note. We have thoroughly revised the “Introduction”, in particular, moving here from section “Results” information about the summertime Atlantic–Eurasian (AEA) teleconnection over northern Eurasia between North Atlantic and the Eurasian continent, based on mid-tropospheric climatic data  [Jianping Li and Chengqing Ruan, 2018, https://doi.org/10.1088/1748-9326/aa9d33]. We also added to the “Introduction” the description of the other Eurasian teleconnection pattern – British–Baikal Corridor (BBC) proposed in [Xu et al, 2019, DOI: https://doi.org/10.1175/JCLI-D-18-0343.1] based on upper tropospheric climatic data. Analyzing the 2019 event, we rely on this information.

  •   High summer rainfall and floods in the continental extratropics are frequently related to regional convective storms.  Additionally, in Siberia, early summer floods are mostly related to the snowmelt (freshet).  This high flow period lasts until June. To link these high flow events to the large-scale atmospheric Rossby waves is an oversimplification. Or, at least, require a thorough justification.

Answer: Thanks for the helpful note. We have thoroughly revised the “Introduction” by adding information and references about regional storms and their relationship with large-scale and synoptic circulation.

Floods in Eastern Siberia at the end of June and July 2019 were caused by heavy rainfall, a number of works have been published on this topic, in particular, the work [Vilfand, 2020, doi:10.1088/1755-1315/606/1/012067]. We added this reference to the article.

Vilfand, R.M.; Kulikova, I.A.; Makarova, M.E. Weather and Climate Features of the Northern Hemisphere in 2019 in the Context of Long-Period Variability. IOP Conference Series: Earth and Environmental Science 2020, 606, 012067, doi:10.1088/1755-1315/606/1/012067. 

Minor comments:

Line 176:  PVU - please, decipher the abbreviation (potential vorticity ??).

Answer: Thanks for the note, we have deciphered the abbreviation: Four synoptic hours (0, 6, 12, 18 UTC) data (potential temperature at the 2 potential vorticity units (PVU) level (dynamic tropopause)

Table 1 column 5:  Please, decipher the meaning of footnote 1 in … (110E)1

Answer: Footnote is a typo. We removed it. Apologize. Thanks for the note.

Line 284: The AEA index for June was most positive in 2019 from 1979.   Please, edit English in this sentence. A possible variant could be:  The June AEA index in 2019 was the most positive since 1979.

Answer: Thanks for the note. We have edited this sentence.

Line 343: … fig.6 instead of fig. fig. 6.

Answer: Yes. Thanks for the note. We have corrected this.

Line 366: … framework … (edit, please). 

Answer: Thanks for the note. We have edited: “The clusters of fires began to form on July 3 and were maintained up to July 15 (fig. 2, blue color).” instead of “The first large clusters of fires began to form on July 3 (fig. 2, blue color). The forest fires were maintained in the first region's framework (blue color) up to July 15”.

Line 503:  Please, prove this suggestion.

Answer: Thanks a lot for the note. We looked for confirmation of this assumption and the data did not confirm it. In this case, it is a horizontal intrusion of cold air from the east. We have changed this part of the text and added an additional figure in Supplementary Materials (Figure S2).

Reviewer 2 Report

Dear authors,

The review comments are provided in the attached pdf file. Thank you. 

Author Response

We are grateful to the Reviewer for the careful reading of the article and for useful comments. According to the comments, we have made the following changes to the text of the article:

General remark:

..However, it focuses too much on wave-breaking events and does not consider other fire-provoking factors.

Answer: Yes, the fire-provoking factors are complex and numerous. It is difficult to concentrate in detail on all factors in one work. Our goal is to analyze the large-scale and synoptic circulation during fires in the summer of 2019.  According to https://www.nasa.gov/image-feature/goddard/2019/huge-wildfires-in-russias-siberian-province-continue and https://tass.ru/proisshestviya/6703544 long-duration forest fires from the end of June to the first half of August in 2019 caused by dry thunderstorms and high air temperature.

… Furthermore, the authors didn’t show how extreme this event is. 

Answer: In summertime 2019, Eastern Siberia (ES) faced the record-breaking flood with peaks at the end of June and the end of July caused by extreme rainfall [https://tass.com/floods-in-irkutsk-region] and extreme, long-duration forest fires from the end of June to the mid of August caused by dry thunderstorms and high air temperature [https://tass.ru/proisshestviya/6703544,  https://www.nasa.gov/image-feature/goddard/2019/huge-wildfires-in-russias-siberian-province-continue , https://go.nasa.gov/3fkG4Dk ]. 

The authors mentioned other factors, such as teleconnection patterns, and dry lightning events, as the cause of the fire events over the study region. But why the authors chose to consider these factors

Answer: AEA teleconnection pattern demonstrates Rossby wave train over northern Eurasia in the summertime [Li J., Ruan C., 2018]. Dry lightning is one of the main causes of natural forest fires. We performed a process-oriented analysis and evaluation of the Rossby wave propagation, their breaking, blocking formation, flood, and forest fire in Siberia in the summer of 2019. Therefore, we use all these factors.

Li J., Ruan C. The North Atlantic–Eurasian teleconnection in summer and its effects on Eurasian climates. Environmental Research Letters 2018, 13:024007. doi: 10.1088/1748-9326/aa9d33

… how dry lightning events are associated with wave-breaking events,... 

Answer: Thanks a lot for the note, we have added information to the text.

Furthermore, precipitation can be formed ahead of anticyclonic RWB, so the statement that anticyclonic RWB leads to fire activity seems to be too hasty. Those types of analyses and clarification are needed before consideration of publication. See the detailed comments below.

Answer: Thanks a lot for the note. Yes, the reviewer is right.  We apologize if our statements sounded categorical. We performed a process-oriented event analysis of summertime circulation in 2019 over Eastern Siberia. There was no extreme precipitation in this event formed ahead of anticyclonic RWB. In the area of the blocking anticyclone, long and intense fires were formed during this event. Our statements apply only to the summer 2019 over Eastern Siberia. 

Major comments:

.. they focus too much on the circulation perspective only. The synoptic circulation pattern indeed helps to provide favorable conditions for fire activity, but this can not explain all fire behaviors.

Answer: Unfortunately, in one work it is impossible to cover all the causes of fires and floods at a sufficiently high level. Therefore, without denying the complexity of the problem, this work is aimed at a detailed analysis of one of the important causes of extreme fires and floods: large-scale and synoptic circulation.

… The circulation change associated with RWB may modulate the fire activity or extreme flooding events, but there is no explanation for why it occurred summer of 2019. The concern and questions increased as reading the manuscript because the authors said these events could be “random” (Line 600). So, more questions arise: Are these just episodic events? Are there distinct climate effects on that? What was the background atmospheric condition? What are the underlying soil moisture, vegetation, temperature, humidity, and wind condition (compared to the climatological mean)? So many questions still remain unanswered.

Answer: We sincerely apologize for not being able to correctly formulate our tasks in the original version of the article. In this work, we associate the event we are studying with the АЕА teleconnection pattern [Li J., Ruan C., 2018, doi: 10.1088/1748-9326/aa9d33]. We consider the events in the European part of this wave train as a possible precursor to extreme events in Siberia. We originally put the AEA overview in the Results section completely in vain. We have thoroughly revised the “Introduction”. We have also added there additional information about other Eurasian summer circulation pattern - British–Baikal Corridor (BBC) proposed in [Xu et al, 2019, DOI: https://doi.org/10.1175/JCLI-D-18-0343.1] and Asian summer monsoon anticyclone (ASMA) [Basha, 2020, doi:10.5194/acp-20-6789-2020]. Our huge apologies and gratitude to the reviewer. Line 600 a completely incorrect statement on our part, we removed it.

Furthermore, the study regions are part of the Asian summer monsoon anticyclone (ASMA) region. The eddy shedding events occur pretty frequently associated with this ASMA. Can you distinguish between ASMA and RWB? Are these RWB events you described the same as the eddy shedding events? Please clarify.

Answer:  Thanks a lot for the new information and useful questions. It seems to us that their depth claims to be a separate study involving model calculations. This article is initially focused on the consideration of west-east relations. But we cannot now fail to mention here possible sources from the south. We adopted these questions as the basis for our further research.

Figures should be self-explanatory. But figures are extremely confusing. Furthermore, the author just described what each plot is, but not what they mean. No interpretation or explanations were given even though they present so complicated, multiple subplots, often duplicated. These often bothered the readers.

Answer: Thanks a lot for the note. We have thoroughly revised the “Introduction” and text (1st (1–19 June) period). We have also moved to Supplementary Materials some of the Figures and added additional Figures to Supplementary Materials to clarify some of the statements.

Minor comments:

As mentioned above, in general, figures are extremely difficult to read and hard to follow. And there are so many figures, some of which are very repeated. Are they all necessary to convey the author’s point? For example, there are so many subplots in Figures 9, 10, 12, and 13. Are they all necessary? If not, I recommend you move most of them into the supplementary materials and leave the most important ones.

Answer: we converted part of the Figures into a video.

As mentioned in the general comments, why summer 2019? Is there any clear difference in the meteorological conditions compared to climatology? What other factors play a role, particularly this year? Then why? There is no consideration of the change in meteorological variables in this time period. I think that should be the starting point, rather than getting to a more complex wave analysis.

Answer: Thanks a lot for the note. We have thoroughly revised the “Introduction”. 

The authors also mentioned many other factors, such as teleconnection patterns, lightning strikes due to dry thunderstorms, etc. But it wasn’t clear how they are linked to RWB events.

Answer: We have thoroughly revised the Introduction and text (1st (1–19 June) period )  and added the text regarding lightning strikes and dry thunderstorms in Section 3.2. “The forest fire ignitions factors.”

Figure 1: What is (a), (b)??? I can't find (a) and (b) in the figure. Figures are extremely confusing.

Answer: Thanks. We've made corrections and noted (a) and (b) in the Figure.

Figure 3: Cannot find any "very significant" WAF pattern except for 6/11, 7/26, 8/25 etc….

Answer:  Thanks a lot for the note. We have thoroughly revised the text about 1st (1–19 June) period.

Figure 4: The continent maps are hard to see/tell…

Answer:  Agree with the reviewer. Too much information is placed on one mao. In order to facilitate the analysis, we have signed longitudes and latitudes. We also made a video and added in Supplementary Materials. We hope that we were able to make map analysis more convenient.

Figure 5: Isn't there any impact from the Asian summer monsoon?

Answer:  Thanks for the interesting question and recommendation. It's worth thinking about. But now, on the basis of Figure 5 alone, we are not entitled to draw unambiguous conclusions about this influence. We mentioned in the revised text of the introduction the possibility of the influence of southern processes, in particular ASMA.

Figure 6: What is the latitude? (cyclonic RWB often develops in the higher latitude)

Answer:  Thanks for the question. Yes, in general, it is. However, works (Bowley, 2019 and Jing, 2018) contain information that in the summer period RWB in Eastern Siberia can occur at the latitudes of Baikal.

Bowley K.A., Gyakum J.R., Atallah E.H. A New Perspective toward Cataloging Northern Hemisphere Rossby Wave Breaking on the Dynamic Tropopause. Monthly Weather Review 2019, 147:409–431. doi: 10.1175/mwr-d-18-0131.1

Jing P., Banerjee S. Rossby Wave Breaking and Isentropic Stratosphere-Troposphere Exchange During 1981-2015 in the Northern Hemisphere. Journal of Geophysical Research: Atmospheres 2018, 123:9011–9025. doi: 10.1029/2018jd028997

Figure 7: Near the surface, PV contours look weird. That is because PV is defined on the isentropic surface (same potential temperature), and it crosses the surface below 320-300 K or so. Clarification about this would help.

Answer:  Thanks for the question. Yes. We think this is due to the influence of complex orography.

Figure 12: This reviewer disagrees with the mark in Figure 12a is CWB. Some increase in wave activity flux is observed, but wave breaking type and shape is too subtle to be determined.

Figure 13: This reviewer disagrees with the mark in Figure 13b, d are AWB…Both types of wave breaking exist here, forming a blocking type of circulation..

Answer:  Thanks for the comments. In order to avoid disagreements in the visual interpretation of maps, an automatic algorithm for identifying breaking was used.

Figure 15: Isn't those "blocking" system you mentioned (formed by two RWB) associated with the Asian summer monsoon anticyclone system? What is the difference?

Answer:  Thank you for your comment. This work focuses on west-east teleconnection. The Asian summer monsoon is a very important and large part of the summer circulation in the Asian part of Eurasia, the existence of which cannot be ignored. But his research requires serious efforts. At some stages of the study, it is convenient to analyze objects separately, for a deeper understanding of each of them before moving on to a comprehensive study of all factors. We are in one of those stages right now. But we remember the monsoon. And we are very grateful to the reviewer for the fact that he sees the problem in a complex way.

Line 531: What is the pnc 3??

Answer:  pnc 3 is a typo. It is fig. 3. Apologize. Thanks for the note.

Line 564: Precipitation is the indicator of breaking….?? What does this mean? Please clarify.

Answer: Thank you very much for the note. Absolutely incorrect statement. We have removed it from the text.

Line 631: What does this mean? No supplementary materials??

Answer: we added Supplementary Materials.

Reviewer 3 Report

Please find my comments in the attached file.

Author Response

We are grateful to the Reviewer for the careful reading of the article and for useful comments. According to the comments, we have made the following changes to the text of the article:

Major issues

[1] One primary distraction is that the English of the article is poor. Numerous problems related to awkward phrasing (some examples: L#54, L#73-74, L#95-96, L#109, L#185, L#201-202, L#206, L#210-211, L#241, L#272, L#279, L#284, L#304, L#351, L#353, L#355, L#375, L#378, L#379, L#381, L#386, L#394, L#442, L#452 (‘decade of July’?), L#456, L#461, L#471, L#499 (should read ‘water vapor content’), L#551, L#576, L#581, L#616-617, L#623, and elsewhere; L# = line number), to improper verb tenses (some examples: L#307, L#375, L#403, L#467, and elsewhere), to poor word choices (some examples: ‘which was existing’ [L#379], ‘..decade of July’ [throughout manuscript], ‘water content’ should read ‘water vapor content’ [L#499-500 and elsewhere], ‘downwelling’ should read ‘subsidence’ [L#477]), and using the plural of a word when the singular is more appropriate (and using the singular of a word when the plural is more appropriate, found throughout the manuscript).

Answer: Thanks a lot for the notes.

L#54 – we changed to “recognized as the most dramatic”

L#73-74  – we removed it when reworking introduction

L#95-96  – we removed it when reworking introduction

L#109 – we changed to “of the trough”

 L#185 – we added “suggested by”

 L#201-202 – we added “located” 

 L#206 –  we added “of”

 L#210-211 – we changed to “We utilized PV-Θ for analysis of air masses transformation and blocking formation in the mid-latitude area”

 L#241 – we changed to “Table 1 is collected the chronology for precipitation (that has led to the floods), forest fires, blocks over the Eastern Siberia, and RWB for Eastern Siberia and western border territory (60°E-120°E).”

L#272 –  we reworked the text of the paragraph

L#279 – we reworked the text of the paragraph

L#284 – we reworked the text of the paragraph

L#304 – we changed to “was cut off and began to move to the southeast”

L#351 – we removed “due to stuck cutoff low”

L#353 – we removed “However, WAF doesn't propagate into Siberia well”

L#355 – we changed to “Fig. 3 shows”

L#375 – is decaying we changed to “..began to decay”

L#378 – “have moved”

L#379 – “ which was existing since” we changed to “has been observed since”

L#381 –  “On 10-14 July (fig. 10c-e), the eastward WAF increased; however, the synoptic” зwe changed to “On 10-14 July (fig. 10c-e), the WAF in front part of the trough increased”

L#386 –  Since July 15, the large forest fire clusters began to form in second and third forest fire regions..” we changed to “Starting from July 15, the large forest fire clusters  in second and  third forest fire regions began to form.. ” 

L#394 – “The condition GHGS>0 was persisted up to July 22.” we changed to “Blocking over Siberia was observed until July 22”.

 L#442 –  “.. in areas difficult to access” we changed to “ in hard-to-reach areas (difficult to assess)”

L#452 –  (‘decade of July’?), - “first and second decade” we changed to “the 1–10 and 10–20 July” 

L#456 – Thunderstorms that occur without significant rainfall on the ground it is called "dry 456 thunderstorms" (DTh) [59, 60]” we changed to “Thunderstorms that occur without significant rainfall on the ground are called "dry thunderstorms" (DTh) [59, 60].”

L#461, The latter is associated with the formation of cutoff filaments with a low potential temperature (or high PV) [59]” we changed to “the instability is associated with the formation of cutoff filaments with a low potential temperature (or high PV)”

 L#471 – waves breaking we changed to “wave breaking”

L#499 – water content’ we changed to “total column water”

L#551 – we removed “However, the model of their relationship was not ordinary”

 L#576 – “occurred nearest blocking decaying” we changed to “ occurred in period of blocking decays”

L#581 – “period was characterized by the cold” we changed to   “the periods of blocking decay

L#616-617 – added a comma before “in the future”  

L#623 – “It needs to consider that some authors have concluded a..change of ..” we changed to “It needs to consider that some authors have concluded the change of”

to improper verb tenses:

 L#307 – “becomes” we changed to “became”

L#375 – we changed to “began to decay”

L#403 –  “can be analyzed” we changed to “ can be identified”

L#467 –  first 467 decade of July, the second decade of July.. we changed to 1-10 и 10-20

and elsewhere.

to poor word choices (some examples: ‘which was existing’ [L#379] -  

‘..decade of July’ [throughout manuscript] - we changed it elsewhere.

‘water content’ should read ‘water vapor content’ [L#499-500 and elsewhere] - “water content” we changed to  “total column water” elsewhere. 

‘downwelling’ should read ‘subsidence’ [L#477]) -  we changed ‘downwelling’ на ‘subsidence’.

[2] Several examples where the authors make a statement of fact that needs either to be proven, reworded to make the statement less absolute, should be left to a future study, or should be deleted from the revised manuscript. Examples,

(a) “…processes associated with the Rossby waves breaking…is the most significant candidate causing both flood and forest fires in a remote region…” (L#72-73)

Answer: Absolutely agree with the comment of the reviewer. We have removed this statement from the text.

(b) “…are more significant in summer than in winter.” (explain why, L#122)

Answer: This statement refers to the work cited in the previous sentence. We have revised the text. 

(c) “These anomalies in remote areas can be explained by teleconnection.” (L#275) The statements related to the NAO and AEA are appropriate for a different study as the proof that they contributed to the events described in the paper require much more analysis than included in the current manuscript. The teleconnections statements distract from the primary message of the important linkages between the large-scale weather wave-breaking features identified and the subsequent periods of significant flooding or wildfires.

Answer: We agree with the reviewer. We have significantly revised this part of the text.

(d) “…was closely related to wave train propagating.” (L#281-282) You must either show this or include a reference that showed this.

Answer: We agree with the reviewer. We have significantly revised this part of the text.

(e) “We have supposed that the extreme AEA teleconnection in June 2019 caused the pressure pattern…” (L#287-288) See comment (c) above, it is recommended this statement be

deleted since it hasn’t been proven in this paper or include a reference in which it was already proven to be true.

Answer: We agree with the reviewer. We have significantly revised this part of the text.

(f) “We suppose two main reasons for the scenario…” (L#321, do you mean to say “hypothesize”?)

Answer: We agree with the reviewer. We have revised this part of the text.

[3] The organization of the manuscript is confusing in places. Examples,

(a) Parts “(a)” [L#90] and “(b)” [L#92] are referenced in the Introduction and then the subsequent paragraphs labeled “(a)” [L#94-106] and “(b)” [L#107-117] contain too much extraneous information unrelated to the original description of the preceding paragraph.

Answer: We agree with the reviewer. We have revised this part of the text.

(b) The earliest part of the Data and Methods (a) Data section [L#132-181] should be converted to a table, with a brief description of the data sources. A reference is needed describing the MOD14/MYD14 dataset. What types of observations go into the GPCC analysis [L#168-175]?

Answer: Thanks for the note. We have added a reference is needed describing the MOD14/MYD14 dataset: 

Giglio, L.; Schroeder, W.; Hall, J.V.; Justice, C.O.MODIS Collection 6 Active Fire Product User’s Guide. Revision B. /  NASA, USA. 2018, 64 p.;

Giglio, L.; Schroeder, W.; Justice, C.O. The collection 6 MODIS active fire detection algorithm and fire products. Remote Sensing of Environment. 2016, 178, 31–41. doi:10.1016/j.rse.2016.02.054.

And we have added the information about GPCC analysis: This set represents ground observations of daily precipitation derived from the quality-controlled stations and interpolated into the grid with a resolution of 1° × 1°

(c) There are currently two (c) sections “Rossby wave breaking” and “Wave activity flux” in the Data and Methods section that should be combined into a single section (c). Please include a bit more discussion of WAF; what it represents and its units.

Answer: We agree with the reviewer. We have revised this part of the text.

(d) The early text of Section 3.1 of the Results section is too disjointed. L#223-235 reads like a figure caption, and the following ‘paragraphs’ (L#237-254) are too abbreviated and in need of more description and details.

Answer: Thanks for the note. We have tried to remedy the situation.

(e) L#244-247 description belongs in the Methodology section.

Answer: Thanks for the note. We have decided not to make changes here

[4] Some figures and tables need improvement to most effectively communicate their information. In some cases, figures can be deleted without negatively affecting the impact of the manuscript. Examples,

(a) Table 1 should be reconstructed as a timeline, comparable to Figure 1. Each column has a slightly different timescale that makes its interpretation difficult. For example, box 24-26 June: (column #1) covers a greater vertical extent in Table 1 than does the 26 June – 3 July box of the final column (column #5), but clearly covers a shorter time period.

Answer:  Thanks for the note. The table was moved to Supplementary.

(b) Figure 1 has colors that are too similar and difficult to differentiate (A0.25, tcw, and WS). Also, please explain the meaning of the blue and yellow arrows labeled ‘1’ and ‘2’ in both the main text and figure caption.

Answer: Thanks for the note. We corrected Figure 1.

(c) Multi-panel Figures 4, 5, 9, 10, 12, and 13 take up too much space in the manuscript. Consider creating animation loops that can be included online in a supplemental or appendix section that can loop through every day of each period. The selected times within each period seem almost arbitrary and are rather discontinuous, making ‘observation’ of the large-scale feature evolution impossible to follow and the text description even more difficult to follow.

Answer: Thanks for the note. Table 1 was placed to Supplementary Materials.

We converted Figures to Video.

(d) Please give the source of gridded data in all figure captions (e.g., Era-Interim).

Answer: Thanks for the note. We have added the source of gridded data in figure captions. 

(e) Please label the dates and times on all figure panels.

Answer: Thanks for the note. We have corrected it.

(f) On what surface are the daily streamlines of Figure 5?

Answer: Thanks for the note. We added: 850 hPa.

(g) Only a single color bar (with units labeled) is needed on Figs. 4, 5, 9, 10, 12, and 13.

Answer: Thanks for the note. We made a video of these Figs.

(h) Does red and blue shading in Figure 7 represent winds blowing into or out of the section? Please clarify in the figure caption.

Answer: Thanks for the note. We added: Red shading - westerly wind (positive) blue -easterly wind (negative)

(i) The color shading in Figure 8 makes it very difficult to differentiate between high and low total column water areas. The high values saturate at too low a threshold.

Answer: the figure is shown so that the maximums are clearly visible, we wanted this. And they did it on purpose.

(j) Show the area borders of Figure 11 on an earlier figure (Fig. 10?) to help the reader know the area covered by the imagery in Fig. 11.

Answer: we have tried. But the area borders made the figure overloaded. In the caption to the figures, we marked the coordinates of the boundaries 60-120E, 30-70N.

(k) The AWBs of Figure 13 are described in the main text, but the AWB outline is missing from panel (e).

Answer: Thanks for the note. An error was found in the dates of events, we have corrected the breaking dates.

(l) consider deleting Figs. 15 and 16 as they add little to support the main conclusions of the study.

Answer: after converting most part of Figures  to video, we decided to leave these  Figs. 15 and 16   in the text.

Minor issues – editorial and technical issues: 

[1] Extreme weather events (EWEs) and extreme precipitation events (EPEs) in the United States were described in Bosart et al. (2017) and Moore et al. (2019) as being preceeded by Anticyclonic Rossby Wave breaking events identified in the upper troposphere. These works must be included in the review of literature found in the Introduction section of the manuscript.

Bosart, L. F., B. J. Moore, J. M. Cordeira, and H. M. Archambault, 2017: Interactions of North Pacific tropical, midlatitude, and polar disturbances resulting in linked extreme weather events

over North America in October 2007. Mon. Wea. Rev., 145, 1245–1273, https://doi.org/10.1175/MWR-D-16-0230.1.

Linkages between Extreme Precipitation Events in the Central and Eastern United States and Rossby Wave Breaking,

Benjamin J. Moore, Daniel Keyser, and Lance F. Bosart

Print Publication: 01 Sep 2019, DOI: https://doi.org/10.1175/MWR-D-19-0047.1.

Answer: Thanks for the note and new information. We included the works in the Introduction.

[2] Please clarify…

(a) “…the precipitation becomes more extreme.” (L#306-307)

Answer:  Thank you, we corrected these phrase.

«On June 23, light precipitation was recorded in the front part of the trough from centers near 75 E. and on June 24, under conditions of cold air intrusion accompanied by the formation of CWB, extreme precipitation was observed»

(b) “…streamline strengthening…” (L#317) What is meant by this and at what level is this occurring?

Answer: Thank you, we added “at 850 hPa” and corrected this phrase.

(c) “…detected two PV breaking:…” (L#384) What detected the PV breaking?

Answer: Thank you, we corrected these phrase.

“For the period July 14 to July 17 was detected two PV contours  breaking:”

(d) “…shows an anomaly of the LS…” (L#451-452) How is a LS anomaly calculated? What is the source of the LS mean in order to calculate a LS anomaly?

Answer: Anomaly in lightning strokes compares to periods 2009-2018. Calculations are made using WWLLN.  We calculated the average for 1-10 and 10-20 July, for 2009 -2018, then we calculated 2019 anomalies relatively these average. 

Lightning strikes were calculated using the WWLLN database. The average for July 1-10 and 10-20 for 2009-2018 was calculated, then the deviation in 2019 from this average was calculated

(e) “…transport of air masses…so-called diabatic heating.” (L#475-476) Horizontal transport of air masses is known as thermal advection, which is distinctly different from diabatic heating in the Thermodynamic Energy Equation.

Answer: thanks, we've corrected this.

(f) “…three factors (diabatic, …, incoming solar radiation),…” (L#482-483) Incoming solar radiation is one form of diabatic heating. This statement is redundant.

Answer: thanks, we've corrected this.

(g) “This led to contrasts increasing…” (L#537) Was this increase in contrasts important for decreasing the vertical stability of the environment or for increasing the baroclinicity of the environment (increasing the strength of the vertical circulation associated with frontogenesis)?

Answer: contrast (baroclinicity) and instability increasing.

(h) “We speculate they can be random.” (L#600) Please be more specific to clarify this statement.

Answer: Thanks, we agree and removed this statement.

Round 2

Reviewer 3 Report

Please find my comments in the attached document.

Author Response

We express our gratitude to the reviewers for their insightful and constructive feedback on our manuscript. Their valuable input has greatly helped us to improve the quality and clarity of our research findings, and we appreciate the time and effort they have dedicated to this review process.

We have made the following changes to the text of the article:

[1] One primary distraction is that the English of the article is poor. <== although this WAS mostly addressed, there are still examples of awkward phrasing in need of improvement;

“Presently the Siberia forest fires have been recognized as the most dramatic phenomenon due to many scientific papers” (L#53-54)

“faced the record-breaking flood” (L#61)

Answer: We have improved the part of sentences, moreover, if the manuscript gets accepted it will be sent to an English editing service (MDPI) to ensure that the language and the grammar of the manuscript are polished to the highest possible standards before publication."

Now we have made the following changes:

lines 53-54 «Presently the Siberia forest fires have 53 been recognized as the most dramatic phenomenon due to many scientific papers» we changed to  “Currently, the Siberian forest fires have been the focus of many scientific papers, recognized as one of the most dramatic phenomena.”

line 61 « In summertime 2019, Eastern Siberia (ES) faced the record-breaking flood with peaks at the end of June and the end of July caused by extreme rainfall …» we changed to  “In the summer of 2019, Eastern Siberia experienced record-breaking floods with peaks at the end of June and the end of July, caused by extreme rainfall…“

line 684 «It needs to consider that some authors have concluded the change of pattern Rossby wave propagation over Eurasia after the mid-90th» we changed to  “It should be noted that some authors have concluded that there has been a change in the pattern of Rossby wave propagation over Eurasia since the mid-1990s.”

line 22 « In the summer of 2019, two types of Rossby wave breaking were observed: cyclonic one, a  wave breaking over Eastern Siberia from the east (110°Е–115°Е), and anticyclonic type, a wave breaking over Eastern Siberia from the west (75°Е–90°Е).» we changed to  “In the summer of 2019, two types of Rossby wave breaking were observed: a cyclonic type, with a wave breaking over Siberia from the east (110°Е–115°Е), and an anticyclonic type, with a wave breaking over Siberia from the west (75°Е–90°Е).”

line 28 «Rossby wave breaking thrice resulted in atmospheric blocking over Eastern Siberia:..» we changed to  «Rossby wave breaking occurred three times, resulting in the formation and maintenance of atmospheric blocking over Eastern Siberia:…».
line 43 «One of the most critical climate indicators is the air temperature and precipitation; droughts, wildfires, and floods in boreal summer are associated with them.» we changed to  “The air temperature and precipitation are among the most critical climate indicators, and they are closely linked to extreme weather events such as droughts, wildfires, and floods during boreal summer.”

line 48 « These changes lead to flood and forest fire occurrence increase[3, 4]. For Russian economics and human health, both fires and floods are extremely dangerous.» we changed to  “These changes have led to an increase in the occurrence of floods and forest fires [3, 4]. Both fires and floods are extremely dangerous for the Russian economy and human health.”

line 60 « … black carbon settles.. » we changed to  “… black carbon settling..

lines 86–89 «The work [29] presents the results of the study of precipitation events over the Selenga river basin and atmospheric blocking over Eurasia for 1979–2017 in July and shows that joint blocking over Europe and the Russian Far East (RFE) contribute to aridity over the southern (Mongolian) part of the Selenga basin and to the precipitation over the northern (Russian) part of the basin.» we changed to «The study presented in [29] analyzed precipitation events in the Selenga river basin and atmospheric blocking over Eurasia during July from 1979 to 2017. The results showed that when there was joint blocking over Europe and the Russian Far East (RFE), it led to aridity over the southern (Mongolian) part of the Selenga basin and increased precipitation over the northern (Russian) part of the basin».

line 128 «It is known that for the boreal forest area, midlatitude circulation is predominant..» we changed to «It is known that midlatitude circulation is predominant for the boreal forest area».

line 141 «Both types of breaking were accompanied of the trough deepening from the sub-Arctic region» we changed to “The trough deepening from the sub-Arctic region was associated with both types of breaking”

lines 153–154 « The aim of this work is a process-oriented analysis and evaluation of the Rossby  wave propagation, their breaking, blocking formation, flood, and forest fire in Siberia in the summer 2019.» we changed to  “The aim of this work is to conduct a process-oriented analysis and evaluation of the Rossby wave propagation, their breaking, blocking formation, floods, and forest fires in Siberia during the summer of 2019.”

line 204 «… for Siberia area» we changed to «…for the Siberia area»

line 215 we removed «that» in «.. criterion that suggested by..»

line 224 «on» instead of «upon»

line 231–232 « It was discovered the area of high CWB frequency located eastward of Lake Baikal for 350 K and PV-Θ » we changed to «It was discovered that the area with a high frequency of CWB is located eastward of Lake Baikal at 350 K»

line 236–237 « For automatically detection of centers and squares of overturning areas, we used the identification technique developed by Barnes and Hartmann ..» we changed to «For the automatic detection of centers and squares of overturning areas, we used the identification technique developed by Barnes and Hartmann».

line 240 «for analysis» we changed to «for the analysis»

line 245 «The wave activity flux (WAF) that indicates a propagating of planetary waves usually can be applied for localizes regions of wave activity sources and sinks» we changed to «The wave activity flux (WAF) that indicates the propagation of planetary waves can usually be used to localize regions of wave activity sources and sinks».

line 273 « is collected the chronology for precipitation » we changed to «is collected the chronology of precipitation»

line 274 we removed «the» before «..Eastern Siberia»

line 276 Fig. 2 shows the clusters of forest fires (FFCs) and everywhere in the text FFCs we removed apostrophe.

line 281 «..on six time-intervals » we changed to «into six time intervals»:

line 346 «Black» we changed to  «The black»

line 313 «We draw attention here to this record-breaking high temperature over Europe as an event preceding the large-scale atmospheric events that we are studying over Siberia and possibly associated with them.» we changed to   « We draw attention to the record-breaking high temperatures over Europe preceding the large-scale atmospheric events that we are studying over Siberia, which could possibly be associated with them »

line 407 «vast» we changed to  «a vast»

line 408 «is depicted» we changed to  «area depicted»

line 411 «The first blocking over Eastern Siberia (fig. 1b) resulted from low PV-Θ air mass intrusion, a cutoff low, and CWB and simultaneous intensification and propagation of WAF from North Atlantic to Siberia» we changed to  «The first blocking over Eastern Siberia (fig. 1b) was caused by the intrusion of a low PV-Θ air mass, a cutoff low, and CWB, as well as the simultaneous intensification and propagation of WAF from the North Atlantic to Siberia».

line 417 « …and were maintained up to July 15» we changed to  «… and persisted until July 15».

line 422 « Here and below red circle with letter A mark the blocking anticyclone position ..» we changed to  «Here and below, a red circle with the letter A marks the position of the blocking anticyclone..»

line 432 «It was due to WAF amplification and extra cold air masses advection in the front part of the anticyclone» we changed to «It was due to the amplification of WAF and the advection of extra cold air masses in the front part of the anticyclone».

line 436 «For the period July 14 to July 17 was detected two PV contours breaking» we changed to «For the period from July 14 to July 17, two PV contours breaking were detected»

line 442 « The onset of the increase of hotspots number and biomass emission and the northward shift of forest fire area (fig.2) in mid-July was simultaneous with the CWB on 16-17 July» we changed to «The increase in the number of hotspots and biomass emissions, as well as the northward shift of the forest fire area (as shown in Figure 2), occurred concurrently with the occurrence of CWB on July 16-17»

line 469 «south» we changed to «southern»

line 472 «Regarding blocking formation, CWBs on 26-28 July are similar to the CWB accompanied the first precipitation period..» we changed to «Regarding the formation of blocking, the CWBs on 26-28 July are similar to the CWB that accompanied the first precipitation period..»

line 477 «for the July 28» we changed to «on July 28»

line 495 «The forest fire ignitions factors» we changed to «The factors contributing to forest fire ignitions»

line 520 « In fig.S2a are shown CAPE (convective amiable potential energy) that characterizing instability in the troposphere.» we changed to «In fig. S2a, CAPE (convective available potential energy) is shown, which characterizes the amount of energy available for convection in the atmosphere».

line 570 « there were » we changed to «there was»

line 584 «Factors affected the extreme precipitation and flood» we changed to «The factors that affected the extreme precipitation and flood» 

line 591 «south» we changed to «southern»

line 602 « In 2019 record-breaking floods and forest fires were observed in Eastern Siberia (ES) from June 24 to August 12»  we changed to «Record-breaking forest fires and floods were observed in Eastern Siberia (ES) between June 24 and August 12 in 2019».

line 626 « Depending on the degree of PV overturning, the rainfall can be quasi-stationary during some days»  we changed to «Depending on the degree of PV overturning, the rainfall can be quasi-stationary for some days».

line 629 «with eastward-moving of the low PV-Θ part of blocking)»  we changed to «with the eastward movement of the low PV-Θ part of the blocking)».

line 637 « Blocking anticyclones is located favorable for forest fire spreading »  we changed to «The location of blocking anticyclones can be favorable for the spreading of forest fires».

line 641 « … to anticyclone, resulting from double breaking from west and east »  we changed to «.. to an anticyclone resulting from double breaking from the west and east».

line 678 «It is not only about long-term frequency but also about a change of features of RWB and blocking formation» we changed to «The issue is not only the long-term frequency but also the changes in the characteristics of RWB and blocking formation»

 [3] The organization of the manuscript is still confusing in places. Examples,

(c) There are currently two (c) sections “Rossby wave breaking” and “Wave activity flux” in the Data and Methods section that should be combined into a single section (c). Please include a bit more discussion of WAF; what it represents and its units. <== this was NOT addressed; if WAF is to remain its own section (d), please give a very brief description of the methodology, what it represents, and its units in order to help the reader with its interpretation and fill out what is currently a VERY brief section

Answer: We combined (c ) and (d) into a single section (c)

(d) The early text of Section 3.1 of the Results section is too disjointed. L#255-267 reads like a figure caption, and the following ‘paragraphs’ (L#269-279) are too abbreviated and in need of more description and details. <== this was NOT addressed; the early portion of Section 3.1 still reads like information that should be contained in a figure caption.

Answer: We decided to include this information in the text and break down the description of Figure 1 into paragraphs to make it easier to read and understand the information presented in the figure. Feedbacks were received about the difficulty of working with the figure when the information was presented as a single continuous text in the caption. Therefore, we divided the description of Figure 1 into paragraphs to facilitate reading and comprehension of the information.

Figures 1, 2, and the table form the basis of the division into periods, which are analyzed throughout the entire article. These characteristics are repeatedly mentioned in the analysis of each individual period. We show them at the beginning to illustrate the general dynamics of the characteristics during the studied summer and to demonstrate the basis on which we divided the periods for further analysis, highlighting the criteria for their differentiation. The text describes the periods from lines 280-287, which is all the information we want to show here without losing coherence in the presentation or getting ahead of ourselves. We made changes to lines 250-254.

The main purpose of these drawings at this stage is to show the general dynamics, while a detailed analysis of them is devoted to the entire subsequent article

We have improved readability of the first part of section 3.1 by adding the following lines 253-256, 258-260, and changing lines 281-283

 [4] Some figures and tables need improvement to most effectively communicate their information. In some cases, figures can be deleted without negatively affecting the impact of the manuscript. Examples,

(b) Figure 1 has colors that are too similar and difficult to differentiate (A0.25, tcw, and WS). Also, please explain the meaning of the blue and yellow arrows of panel (b) in both the main text and figure caption. <== this was NOT addressed

Answer: Thanks, we added «Light-blue vertical line – high precipitation events, yellow vertical line – start and finish of forest fire period.» into fig.1 capture.

(i) The color shading in Figure 8 makes it very difficult to differentiate between high and low total column water areas. The high values saturate at too low a threshold. <== this was NOT addressed; I cannot see from the current color scale if the total columnar water vapor maximum toward the southeast is greater in panel (c) or in panel (d)

Answer: Thanks, we corrected figure 8

Bosart, L. F., B. J. Moore, J. M. Cordeira, and H. M. Archambault, 2017: Interactions of North Pacific tropical, midlatitude, and polar disturbances resulting in linked extreme weather events

over North America in October 2007. Mon. Wea. Rev., 145, 1245–1273, https://doi.org/10.1175/MWR-D-16-0230.1.

Moore is mis-spelled (“Moor”, L#84) in the text

Answer: Thanks, we fixed it

[2] Please clarify…

(g) “This led to contrasts increasing…” (L#599-600) Was this increase in contrasts important for decreasing the vertical stability of the environment or for increasing the baroclinicity of the environment (increasing the strength of the vertical circulation associated with frontogenesis)? Please answer this question for the reader in the body of the manuscript. It is unclear in its current form.

Answer: we clarified: The contrast between warm moist air from the East Asian summer monsoon area (video V2) and cold arctic air from the Kara Sea  (video V1 and video V5) , which was involved in the southern part of Eastern Siberia during the cyclonic Rossby wave breaking, led to an increase in the strength of the vertical circulation associated with frontogenesis. This resulted in extreme precipitation, contributing to the 2019 Siberian summer anomaly.
